# Red edge effect and chromoselective photocatalysis with amorphous covalent triazine-based frameworks

Yajun Zou[1,2], Sara Abednatanzi[3], Parviz Gohari Derakhshandeh[3], Stefano Mazzanti [1], Christoph M. Schüßlbauer [4], Daniel Cruz[5,6], Pascal Van Der Voort [3], Jian-Wen Shi [2], Markus Antonietti [1], Dirk M. Guldi [4] & Aleksandr Savateev [1✉]

Chromoselective photocatalysis offers an intriguing opportunity to enable a specific reaction pathway out of a potentially possible multiplicity for a given substrate by using a sensitizer that converts the energy of incident photon into the redox potential of the corresponding magnitude. Several sensitizers possessing different discrete redox potentials (high/low) upon excitation with photons of specific wavelength (short/long) have been reported. Herein, we report design of molecular structures of two-dimensional amorphous covalent triazine-based frameworks (CTFs) possessing intraband states close to the valence band with strong red edge effect (REE). REE enables generation of a continuum of excited sites characterized by their own redox potentials, with the magnitude proportional to the wavelength of incident photons. Separation of charge carriers in such materials depends strongly on the wavelength of incident light and is the primary parameter that defines efficacy of the materials in photocatalytic bromination of electron rich aromatic compounds. In dual Ni-photocatalysis, excitation of electrons from the intraband states to the conduction band of the CTF with 625 nm photons enables selective formation of C–N cross-coupling products from arylhalides and pyrrolidine, while an undesirable dehalogenation process is completely suppressed.

---

[1] Department of Colloid Chemistry, Max Planck Institute of Colloids and Interfaces, 14476 Potsdam, Germany. [2] State Key Laboratory of Electrical Insulation and Power Equipment, Center of Nanomaterials for Renewable Energy, School of Electrical Engineering, Xi'an Jiaotong University, Xi'an 710049, China. [3] Center for Ordered Materials, Organometallics and Catalysis, Ghent University, 9000 Gent, Belgium. [4] Department of Chemistry and Pharmacy Interdisciplinary Center for Molecular Materials (ICMM), Friedrich-Alexander University Erlangen-Nürnberg, 91058 Erlangen, Germany. [5] Department of Inorganic Chemistry, Fritz-Haber-Institut der Max-Planck-Gesellschaft, Berlin 14195, Germany. [6] Department of Heterogeneous Reactions, Max Planck Institute for Chemical Energy Conversion, Mülheiman der Ruhr 45470, Germany. ✉email: Oleksandr.savatieiev@mpikg.mpg.de

Semiconductor photoredox catalysis opens up the possibility of realizing complex multistep chemical reactions under mild conditions[1,2]. In the past few years, a booming development in conjugated polymers, starting from polymeric carbon nitride materials composed of heptazine units[3–5], and covalent organic frameworks (COFs) with repeating molecular units linked by strong covalent bonds, is witnessed[6–9]. Typical controlled synthetic routes toward COFs involve integrating building blocks with appropriate linkage, constructing a π-conjugated system, and introducing electron donor–acceptor units into the framework[10]. This allows precise modulation of light absorption, band positions, as well as intramolecular/intermolecular charge transport/separation behaviors. Vyas et al.[11] reported, for example, a strategy for building an azine-linked COF platform with adjustable electronic and structural properties for photocatalytic $H_2$ production. Huang et al.[12] demonstrated that the charge separation in a covalent triazine framework is promoted by altering electron donor–acceptor moieties, which leads to the enhancement of activity in photoredox catalysis. Enhancement of COFs photocatalytic performance through inclusion of electron rich and electron poor conjugates is a viable strategy, but application of such materials is limited to non-oxidizing environment and pH-values close to neutral[13]. One distinct advantage of COFs is their regular nature, which enables the materials to serve as an ideal platform for structure engineering on a molecular level[14]. The feature of porosity and optoelectronic properties determined by the molecular structure can thus be finely tuned to attain desired functionality[15–17]. The tunability, along with their extended π-conjugated framework, regular pore structure, and high surface area, make COFs attractive alternatives to inorganic semiconductor photocatalysts[18].

Reduction and oxidation power of photocatalysts play a crucial role in enabling a specific reaction. Ultimately, the chemical composition of a semiconductor determines the potential of conduction band (CB) and valence band (VB), which in turn define the driving force for photoinduced electron transfer (PET) in photocatalytic applications[19]. Intriguing alternative to the abovementioned common strategies to adjust redox power of the photocatalytic system is chromoselective photocatalysis, which has been proposed by Ghosh and König[20]. In their approach, illumination of a photocatalytic system composed of Rhodamine 6 G, a homogeneous sensitizer, and electron donor with green or blue photons gives either moderately (–1 V) or highly reductive (–2.4 V vs SCE) species, respectively. In semiconductor carbon nitride photocatalysis, Kroutil et al.[21] and our group[22] achieved various oxidation potentials of the excited state by selective excitation of n–π* or π–π* transitions. Overall, in chromoselective catalysis, one specific reaction pathway (out of multiple possible) is enabled by the photon of specific wavelength[23–25]. Schematically a general principle behind the chromoselective photoredox catalysis enabled by photons with shorter and longer wavelength is depicted in Fig. 1a.

Despite plentiful reports on synthesis of crystalline COFs for photocatalytic $H_2$ evolution[15,26,27], amorphous COFs (hereafter referred to as CTFs) offer a unique platform for designing materials with strong red edge effect (REE)[28]. REE conserves the information about the energy of incident photons in the emission spectra, which is very different from majority of fluorophores in solution and crystalline semiconductors (Fig. 1b)[29]. REE was solely studied from a fundamental perspective using, for example, graphene oxide[30,31]. To the best of our knowledge, using REE in the area of photocatalysis remains to this date rather elusive. Materials with REE break the Kasha's rule due to the existence of distribution of excited states on the timescale potentially sufficient for electron transfer (Fig. 1c). As a result, using incident photons with specific wavelength would yield a continuum of excited sites

with their own energies and redox potentials that may be used in chromoselective organic photoredox catalysis (Fig. 1b, c).

Herein, we present two-dimensional (2D) amorphous covalent triazine-based frameworks possessing strong REE. Nitrogen substitution in the CTFs peripheral aryl rings modulates porosity, optical gaps, separation of charge carriers and density of intra-band states. CTFs allow for enhanced visible-light-driven bromination of aromatic compounds using HBr and $O_2$, even under low energetic 625 nm photons. In dual Ni-photocatalysis, red light is essential to suppress completely the undesirable dehalogenation reaction and obtain selectively the product of C–N cross coupling between arylhalides and pyrrolidine. Structure of the synthesized CTFs and their activity in these two photocatalytic transformations is rationalized based on the spectroscopic study. High-energy π–π* and low-energy n–π* transitions in the CTFs are responsible for the observed selectivity.

## Results

**Structural characterization.** Two CTFs were synthesized by the polycondensation of benzene-1,4-dicarboximidamide and different aldehydes (isophthalaldehyde and 2,6-pyridinedicarboxaldehyde) with procedures as described in ESI, denoted as PHT and PYT, respectively (Fig. 1d, e). By varying the precursor aldehydes, CTFs with two distinct chemical structures are expected to be obtained, where the C-H moiety in the peripheral aryl rings in PHT is substituted with one nitrogen atom in PYT (see the extended chemical structures in Supplementary Fig. 1). This designed structure endows PYT with different electron donor–acceptor domains compared with PHT (marked in different color in Fig. 1d, e). The CTFs gave X-ray diffraction (XRD) patterns with rather broad peaks that indicate amorphous structures (Supplementary Fig. 2). As in most cases of CTFs synthesis[6,32,33], the disorder and amorphous phases formed during the condensation are likely responsible for the limited crystallinity. Both PHT and PYT show a main peak at around 25.1°, which could be ascribed to the π–π interlayer stacking motif corresponding to a distance of 0.354 nm. The additional peak at 18.0° in PYT may be assigned to the ultra-micropores (< 0.5 nm) in the CTF structure[34].

Additional information on the chemical structure of the CTFs was further gained through Fourier-transform infrared (FT-IR) spectroscopy, X-ray photoelectron spectroscopy (XPS), and energy-dispersive X-ray analysis. The FT-IR spectrum of PHT shows two characteristic vibrations at 1514 and 1349 $cm^{-1}$ (Fig. 2a), which can be attributed to the aromatic C–N stretching and breathing modes in the triazine unit, respectively[33]. This result clearly supports formation of triazine linkages. The slight shift of the absorption band due to C–N stretching vibration in PYT indicates the change in C-N bond interaction due to the introduction of pyridinic nitrogen.

In high-resolution XPS spectra (Fig. 2b–d), the C 1 s peak at the lower binding energy is assigned to the adventitious carbon (C–C) along with a contribution from the aromatic carbon (C=C) in the CTF framework, and is calibrated at 284.8 eV[11]. The other three C 1 s signals with binding energies at 285.9, 287.0, and 288.4 eV in PHT correspond to carbon atoms in C=N, C–O, and C–N chemical states, respectively[35,36]. The simultaneously increased content of C=N and C–N groups in PYT implies formation of pyridine rings in the framework (see the integrated area ratios of the deconvoluted peaks in Supplementary Table 1). The large N 1 s peak centered at 398.7 eV is attributed to the pyridine nitrogen (C=N–C) from the triazine ring[37], which is also present in the pyridine ring of PYT. The other peak at around 400.0 eV is assigned to the $NH_x$ groups from the edges of the frameworks[38]. The high-resolution O 1 s spectrum suggests

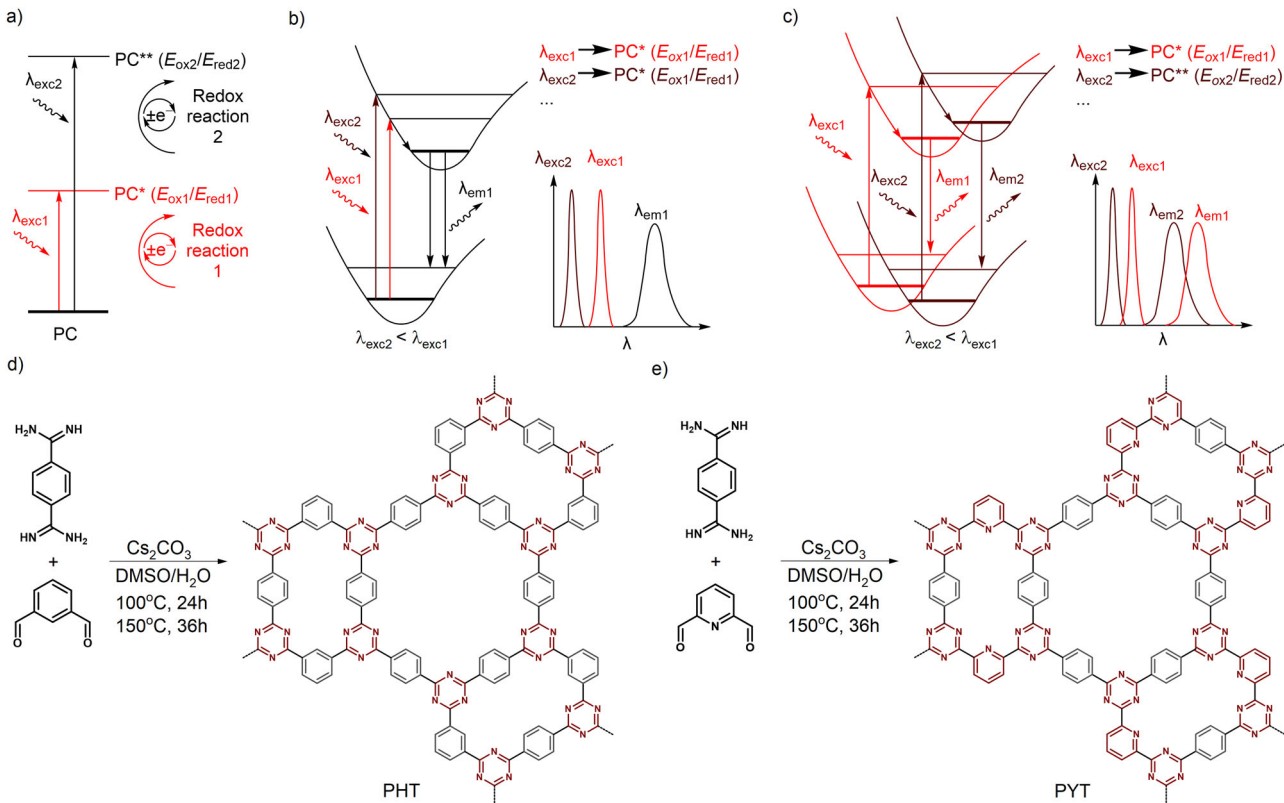

**Fig. 1 The concept of the article. a** A concept of chromoselective photocatalysis. Low ($\lambda_{exc1}$) and high ($\lambda_{exc2}$) energy photons give photocatalyst (PC) excited states PC* and PC** with energies $E_1$ and $E_2$ and the corresponding reduction/oxidation potentials. **b** Illustration of Franck-Condon principle for a fluorophore in which the emission wavelength ($\lambda_{em}$) is independent of excitation wavelength ($\lambda_{exc}$). Low- and high-energy photons generate excited state with the same energy and redox potentials. Position of maximum in the emission spectrum does not depend on $\lambda_{exc}$. **c** Same for a molecule in which emission wavelength ($\lambda_{em}$) depends on excitation wavelength ($\lambda_{exc}$). Selection of the wavelength of incident photons yields PC excited states with their own energies and redox potentials. Position of the maximum in the emission spectrum depends on $\lambda_{exc}$ (red edge effect). **d**, **e** Synthesis of PHT and PYT from the monomers. Electron-rich and electron-deficient domains in the CTFs are marked in gray and red, respectively.

that the surface of the CTFs contains C=O and C−O moieties, with binding energies at 531.7 and 533.1 eV in PHT, respectively[39], which have been incorporated as a result of partial terephtalamidine hydrolysis under alkaline conditions. An increased proportion of the C−O chemical state was observed in PYT, indicating more functional groups with oxygen singly bound to carbon (e.g., C–OH, C–O–C, and C–OOH). Note that the C $1s$ and O $1s$ peaks in PYT appear at the higher energy compared with PHT, indicating that the electron density of the C and O atoms has been shifted toward the electron-deficient pyridine ring. The measured elemental composition of PYT and PHT agrees with the theoretical values (Supplementary Table 2).

Incorporation of heteroatoms into the skeleton of COFs can significantly affect pore structure[40]. The morphologies of the CTFs probed by scanning electron microscope (SEM) show that the materials consist of aggregated nanoparticles with the diameter >500 nm (Supplementary Fig. 3). The transmission electron microscopy (TEM) confirmed amorphous structure of the synthesized CTFs, which is in agreement with the PXRD patterns or could be a result of the CTFs instability under the electron beam (Supplementary Fig. 4)[11,15]. Compared with PHT, smaller and loosely packed aggregates were observed for PYT, implying a more porous structure.

The porosity of the CTFs was further investigated by N$_2$ absorption–desorption measurements. The Brunauer–Emmett–Teller (BET) surface area (S$_{BET}$) of PYT was found to be 104 m$^2$ g$^{-1}$, with a pore volume of 0.36 cm$^3$ g$^{-1}$, higher than those of PHT (27 m$^2$ g$^{-1}$ and 0.06 cm$^3$ g$^{-1}$). Lower surfaces of the CTFs might stem form

staking neighboring layers in AB fashion[41–43]. However, the pore volume of PYT is comparable to that of the CTFs reported in the previous studies[9,44,45]. Pore size distribution was analyzed based on the quenched solid density functional theory model. Pores with the similar diameter of around 2.0 nm were determined for both of the CTFs (Supplementary Fig. 5), in good agreement with the predicted width (1.9 nm) of the largest in-plane pores created by the layers arranged in AA stacking (Supplementary Fig. 1). By virtue of the nanocrystalline nature of CTFs, a precise determination of the stacking type, that is, AA versus AB, is impossible. Larger mesopores in the range of 5.0–16.0 nm observed for PYT are likely associated with the missing columnar π-walls – faults that occur during the synthesis. These nanopores can offer an essential space and can potentially act as nanoreactors for catalytic reactions. Increased S$_{BET}$ and mesoporous structure of PYT are expected to be more favorable for catalytic applications. Thermal gravimetric analysis measurements performed under a N$_2$ atmosphere reveal that PHT and PYT are stable up to 300 °C without significant loss of mass (around 10%) (Supplementary Fig. 6).

**Optical properties and excited-state dynamics.** The light absorption/emission of COFs is strongly dependent on the electron density distribution along the molecule as well as the delocalization of the π-electrons throughout a π-conjugated backbone[46]. The rational introduction of substituents into a conjugated organic molecule, therefore, is a practical means to manipulate the optical properties. As shown in Fig. 3a, the

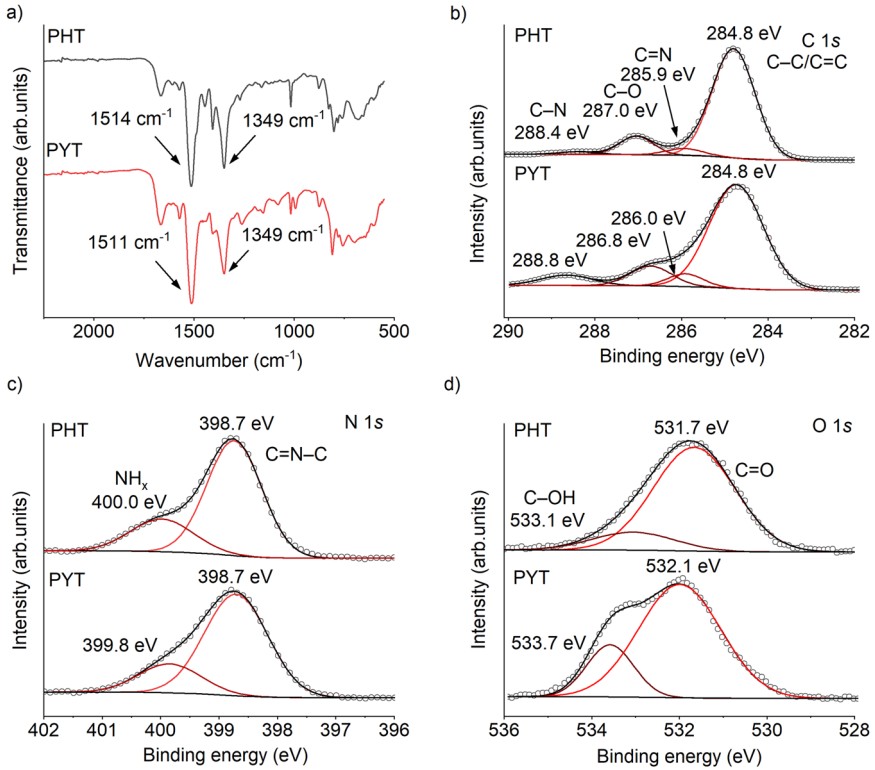

**Fig. 2 Structural characterization of CTFs. a** FT-IR spectra of PHT and PYT. **b** C 1s. **c** N 1s. **d** O 1s XPS spectra of PHT and PYT.

intrinsic absorption bands of the CTFs in the lower λ range are the result of π–π* electron transitions in the conjugated aromatic system[47]. The absorption bands located at higher λ are assigned to n–π* electron transitions involving lone pairs of the nitrogen atoms of the triazine units[48]. PYT shows a notably enhanced n–π* absorption up to near IR (nIR) region compared with PHT, which is rationalized by the additional conjugation with the pyridinic nitrogen. This is consistent with the color change, from yellowish to brownish (inset in Fig. 3a). A narrower optical band gap ($E_g$) was observed for PYT (2.50 eV) in comparison to PHT (2.72 eV) (see corresponding Tauc plots in Supplementary Fig. 7b). We attribute the shrinkage of the band gap to an increased planarity of the framework due to the nitrogen sub-stitution, which leads to a higher level of delocalization[11,49,50]. The VB levels in CTFs were determined from photoelectron spectra (Supplementary Fig. 8). The CB levels in CTFs were calculated by adding the $E_g$ values derived from Tauc plots to the VB levels and are −3.51 and −2.93 eV for PHT and PYT, respectively (Fig. 3b). From Mott–Schottky plots (Supplementary Fig. 9), flat band potentials ($E_{FB}$) were determined as 0.17 V and 0.60 V below the CB edges in the CTFs.

In the photoluminescence (PL) spectra of PHT and PYT, the position and intensity of emission maxima depend on the excitation wavelength, which is known as red-edge effect (REE) (Fig. 3c, d)[28]. REE has been observed in numerous amorphous carbon-based rigid systems, such as graphene oxide and carbon quantum dots[30,51]. REE is related to the existence of the excited-state distribution of fluorophores on their interaction energy with the environment. In other words, in such amorphous systems relaxation of the excited state from higher to zero vibrational level occurs on the timescale longer than the emission of photons via fluorescence.

The chemical structure of the synthesized CTFs affects the REE. Thus, in the PL spectrum of PHT, the emission peak progressively shifts toward longer wavelengths upon increasing the excitation wavelength. The intensity of the PL signal maximizes at 527 nm upon excitation at 450 nm. In the PL spectrum of PYT, the fluorescence maximum also shifts toward longer wavelength along with the increase of $\lambda_{exc}$. But intensity progressively increases along with $\lambda_{exc}$ and maximizes at 715 nm upon excitation at 650 nm. These results agree well with the extended absorption edge of PYT, suggesting that the n–π* absorption band contributes to the fluorescence. Nanocrystaline CTF, PYTnc that was synthesized by polycondensation of the Schiff base of pyridine-2,6-dicarboxaldehyde and terephthalami-dine dihydrochloride (Supplementary Fig. 10), also demonstrates REE in the steady-state fluorescence spectra (Supplementary Note 1).

Fluorescence intensity of PYT is significantly quenched compared with PHT upon excitation at 450 nm, indicating a suppressed radiative charge carriers recombination (Supplementary Fig. 11)[52]. A possible explanation is that the larger electron-acceptor domains in PYT comprising triazine and pyridine units with strong electron-withdrawing functionality render the photogenerated charge carrier separation more favorable. Similar phenomenon was observed in a designed electron-transfer system composed of a cascade of electron acceptors[52].

To further probe the charge transfer process, time-resolved (TR)-PL spectra of the CTFs were recorded at three $\lambda_{exc} = 375$, 470, and 640 nm, while monitoring the decay of photons with the wavelength in the range $\lambda_{em} = 400$–900 nm (Fig. 3e, Supplementary Fig. 12). The decay profiles were fitted using a multi-exponential decay model and $\bar{\tau}$ (amplitude average lifetime) is reported (see the calculation method in the Supporting Information)[53]. The fitting parameters are summarized in Supplementary Table 3 and an example of the decay curve with instrument response function is shown in Supplementary Fig. 13. Taking into account low fluorescence quantum yield (<0.1%), fluorescence spectroscopy does not characterize comprehensively the dynamics of the CTFs excited states. However, lifetimes

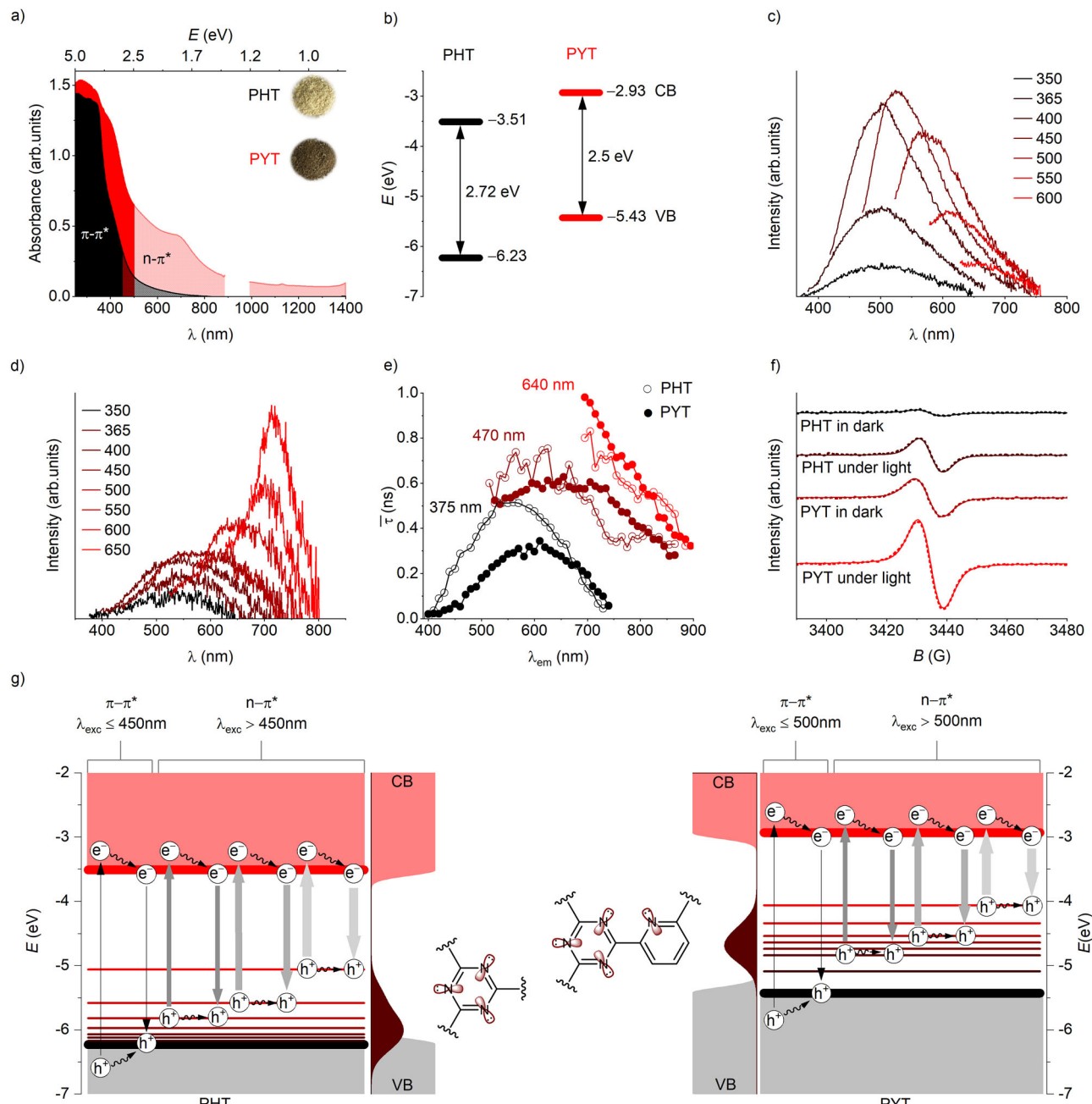

**Fig. 3 Spectroscopic characterization of CTFs. a** Optical absorbance spectra of PHT and PYT. Some discontinuous data points caused by the change in detector at ~900 nm were removed. **b** Band structure of PHT and PYT. **c** PL spectra of PHT. $\lambda_{exc}$ (in nm) is shown. **d** PL spectra of PYT. $\lambda_{exc}$ (in nm) is shown. **e** Amplitude average lifetime ($\bar{\tau}$) of PHT and PYT obtained at $\lambda_{exc}$ = 375, 470, and 640 nm and $\lambda_{em}$ = 400–900 nm. **f** EPR spectra of PHT and PYT at room temperature in the dark and under light ($\lambda_{exc}$ = 455 nm) irradiation (~40 min). Dashed and solid curves show the raw and fitted data, respectively. **g** Band structure of PHT and PYT. Vertical plain lines indicate separation of holes and electrons and their radiative recombination; horizontal and diagonal wavy lines – migration of holes and electrons to surface states, respectively. Schematically thickness of the vertical lines illustrates the rate of photogenerated charge carriers separation (in relation to **e**): the thinner the line the faster is the separation of charge carriers. Schematic representation of the density of states of emissive intraband states based on the data of steady-state PL (**c** and **d**). Nonbonding N orbitals in the structure of PHT and PYT that contribute to the intraband states are schematically shown by the fragments of triazine and triazine-pyridine-diyl linkers.

(≤1 ns) extracted from fluorescence decay curves correspond to the time of charge carriers separation. Analysis of data shown in Fig. 3e enables us to make the following conclusions related to the excited state dynamics in the CTFs. REE strongly affects charge separation in the CTFs[48,54]. Thus, a faster separation of photogenerated charges is achieved upon excitation with more energetic photons such as $\lambda_{exc}$ = 375 nm compared with 470 nm or 640 nm (see also Supplementary Note 2 for additional explanation of TR-PL data). Upon excitation at 375 nm, PYT shows shorter $\bar{\tau}$, and hence more efficient charge separation, compared with PHT. The situation is opposite upon excitation of CTFs at 640 nm, which leads to higher activity of PHT compared with PYT in the photocatalysis under red light (see below) (excited-state investigations by transient absorption spectroscopy revealed a complex deactivation behavior on a ps to ns timescale together with excited-state dynamics exceeding lifetimes >50 µs

(Supplementary Note 3). Possible rationales for such long-lived states are the formation of deep traps or free charge carriers population long-lived surface states. The latter might be beneficial for application in photocatalysis that is based on energy transfer. Recently, the dependence of charge separation on excitation wavelength in carbon nitrides possessing nitrogen vacancies close to VB has been investigated[55].

Furthermore, the CTFs were investigated by electron paramagnetic resonance (EPR) spectroscopy (Fig. 3f). In the absence of light, both CTFs show the presence of stable organic species typical for polymeric networks with extended π-conjugation[56]. Specific concentration of free radicals in PYT was determined to be $1.24 \times 10^{17} g^{-1}$, while in PHT it is $2.35 \times 10^{16} g^{-1}$. We relate the higher concentration of free radicals in PYT to higher nitrogen content that is due to pyridinic moieties capable to stabilize free radicals in the π-conjugated framework[57]. Upon cooling to 90 K, the amplitude of the signal in EPR spectra of both CTFs increases, indicating a paramagnetic ground state (Supplementary Fig. 14)[58]. In situ EPR spectra of CTFs recorded upon samples irradiation with visible light show gradual increase of signal amplitude. In case of PHT, concentration of free radicals increases ca. 3.5 times and reaches saturation within first 5 min of irradiation, while PYT shows gradual accumulation of free radicals over the time and after 40 min increases by ca. 2.2 times (Supplementary Fig. 15). This clear difference suggests an inhibited recombination and hence a prolonged lifetime of photoexcited charge carriers in PYT.

The results of spectroscopic study are summarized in Fig. 3g. Thus, excitation of the materials with photons, ≤450 nm for PHT (optical gap 2.72 eV) and ≤500 nm for PYT (optical gap 2.50 eV), enables π-π* transitions that lead to the formation of 'hot' electrons and holes followed by their migration in the π-conjugated structure of the materials to less energetic surface states, which explains Stokes shift of ca. 50 nm (at $\lambda_{exc}$ 365 nm). Excitation of CTFs with photons of lower energy, >450 nm for PHT and >500 nm for PYT, enables low energetic n-π* transitions. Amorphous (and nanocrystalline in case of PYTnc) structure of CTFs perturbs the electron transfer between nonbonding orbitals composed of nitrogen atoms in triazine (in case of PHT) and in triazine/pyridine (in case of PYT). Therefore, for n-π* transitions the holes remain localized at nitrogen atoms. The REE onset determined from steady-state emission spectra correlates with the $E_g$ (defined by π-π* transitions) of the CTFs – the narrower energy gap the more red-shifted the onset of REE (Supplementary Fig. 16).

Analysis of absorption and steady-state emission spectra also suggests that the density of the intraband states in the CTFs is different. Considering that the strongest fluorescence for PHT is registered upon excitation at 450 nm (Fig. 3c), while the material does not absorb photons with λ > 800 nm (Fig. 3a), intraband states gradually evolve in the middle (from ca. −5.1 eV vs vacuum level) of the PHT band gap and merge with the VB. In PYT, introduction of nitrogen atoms leads to the gradual evolution of intraband states starting from −4.1 eV as evidenced by absorption of the material in nIR (Fig. 3a), while steady-state PL spectra (Fig. 3d) indicate localization of the states in the middle of the band at ca. -4.5 eV. Overall, incident photons generate a population of oxidative states with the energy (versus vacuum level) in the range, relation (1):

$$E_{IBSO} < E_{ox} < \left( E_{CB} - \frac{1240}{\lambda_{exc}} \right)$$

where $E_{CB}$ represents the energy of the CB minimum versus vacuum level, eV; $E_{IBSO}$ represents energy of the intraband states onset versus vacuum level, eV; and $\lambda_{exc}$ represents wavelength of

**Table 1 Screening of reaction conditions.**

| Entry | Photocatalyst | Time (h) | Light | Yield (%)[a] |
|---|---|---|---|---|
| 1 | PYT (4 mg) | 4 | 468 nm[b] | 82 |
| 2 | PHT (4 mg) | 4 | 468 nm[b] | 7 |
| 3 | PYTnc (4 mg) | 4 | 468 nm[b] | 100 |
| 4 | PYT (4 mg) | 24 | 468 nm[b] | 99 |
| 5 | PHT (4 mg) | 24 | 468 nm[b] | 56 |
| 6 | - | 24 | 468 nm[b] | 2 |
| 7 | PYT (4 mg) | 24 | – | 7 |
| 8[c] | PYT (4 mg) | 24 | 468 nm[b] | 97 |
| 9[d] | PYT (4 mg) | 24 | 468 nm[b] | 99 |
| 10[e] | PYT (4 mg) | 24 | 468 nm[b] | 98 |
| 11[f] | PYT (4 mg) | 24 | 468 nm[b] | 96 |
| 12[g] | PYT (4 mg) | 24 | 468 nm[b] | 95 |
| 13 | PHT (4 mg) | 24 | 625 nm[h] | 27 |
| 14 | PYT (4 mg) | 24 | 625 nm[h] | 12 |
| 15 | PYTnc (4 mg) | 24 | 625 nm[h] | 75 |

Reaction conditions: photocatalyst (4 mg); anisole (0.02 mmol); HBr (0.1 mL, 48 wt.%); MeCN (0.5 mL); electron scavenger – $O_2$; at room temperature.
[a]Yields estimated by ¹H NMR with 1,4-dinitrobenzene as internal standard. Exemplary NMR spectrum of the reaction mixture is shown in Supplementary Fig. 22.
[b]Blue LED module 1 (468 nm, 14 mW cm⁻²).
[c]Reaction with 0.02 mL of HBr (48 wt.%).
[d]Reaction with 0.05 mL of HBr (48 wt.%).
[e]Second run.
[f]Third run.
[g]Fourth run.
[h]Red LED module (625 nm, 302 mW cm⁻²).

incident photon, nm. The relation is valid as long as the energy of incident photons is sufficient for excitation of either π-π* or n-π* transitions and defined by the Planck–Einstein relation (2):

$$\frac{1240}{\lambda_{exc}} > (E_{CB} - E_{IBSO})$$

Analysis of the spectroscopic data clearly indicates that despite the fact that chemical structures of PYT and PHT differ mainly in the nitrogen content, such differences have profound influence on materials properties – pore structure, optical gaps, separation of photogenerated charge carriers, and density of intraband states. Such differences inspired us to explore these materials as photocatalysts in organic syntheses.

**Application in oxidative halogenation.** Here, we chose photocatalytic oxidative bromination of aromatic compounds, since this method offers an alternative strategy for the synthesis of halogenated hydrocarbons – important intermediates in organic synthesis[59,60]. Anisole was used as a model substrate, while HBr and $O_2$ were used as the bromide source and electron-scavenger, respectively. In all experiments, 4-bromoanisole **2a** was obtained as the only product (Table 1). An 82% yield of the product was observed in the presence of PYT after 4 h of irradiation (entry 1), whereas PHT only gave a 7% yield (entry 2). Despite surface area of PYT is 4 times higher compared with PHT, such difference cannot account entirely for 12 times higher yield of anisole in the model reaction. PYTnc gave 4-bromoanisole as the only product (entry 3). The yield was increased to 99% and 56% with PYT (entry 4) and PHT (entry 5), respectively, when the irradiation time was extended to 24 h. Low yields (2%–7%) of product were

**Fig. 4 Scope of the photocatalytic bromination of aromatic compounds using PYT as a photocatalyst.** Reaction conditions: substrate (0.6 mmol); HBr (0.6 mL, 48 wt.%); MeCN (3 mL); PYT (4 mg); electron scavenger – $O_2$; at room temperature. [a]Isolated yields. [b]Yields determined by [1]H NMR with 1,4-dinitrobenzene as internal standard. NMR spectra of reaction mixture are shown in Supplementary Figs. S24–S29. [c]Reaction conditions: EDOT (0.2 mmol); KBr (119 mg, 1 mmol); DMSO-$d_6$:$H_2O$ (1 mL, 9:1); PYT (12 mg); $O_2$ (1 bar); at room temperature[2]. [d]Blue LED module 2 (461 nm, 101 mW cm$^{-2}$). [e]White LED module (400–760 nm, 203 mW cm$^{-2}$).

obtained without photocatalyst (entry 6) and without light irradiation (entry 7). Even with lower amount of HBr (entry 8 and 9), we were able to observe a nearly 99% yield with PYT. Addition of variable quantities of $H_2O_2$ into the reaction mixture gave 4-bromoanisole with lower yield, which may be explained by overoxidation of the product under such conditions (Supplementary Table 4). These results obviously point at advantages of photocatalytic generation of $H_2O_2$ in situ. Despite quite strong acidic reaction conditions, no evident decay in the activity (entries 10–12) and no significant structural difference (FT-IR, DRUV–vis, steady state, and TR PL spectra in Supplementary Figs. 17–20 and C, N, H elemental analysis in Supplementary Table 5) could be detected after four cycles, manifesting the extraordinary stability of PYT. Results of control experiments to check for "leaching" suggest the lack of any organics derived from PYT, which are soluble in acetonitrile and which could act as sensitizers in the oxidative bromination of anisole (Supplementary Table 6). The apparent quantum yield of the reaction was further measured and determined to be 0.05% for PHT and 0.18% for PYT under blue light (455 nm), and 0.03% for PHT and 0.05% for PYT under green light (530 nm), respectively (Supplementary Fig. 21).

Inspired by the superior activity of PYT in the visible-light-driven bromination of anisole, we proceeded to examine the scope of the reaction. A series of substituted aromatics were selected as the substrates (Fig. 4). In these experiments, a higher amount of substrate (0.6 mmol) was used to serve the purpose of examining the scalability of the reaction. 4-bromoanisole **2a** was obtained in excellent isolated yields (97%). The reaction was further scaled up to 6 mmol of **1a**, while keeping the reaction time unchanged (48 h) via increasing the photon flux proportionally to the amount of the substrate (Supplementary Fig. 23). In this case, the yield of **2a** was 87%. 4-bromo-2-chloroanisole **2b** was obtained as the only product using 2-chloroanisole **1b** as the substrate, with a good isolated yield of 80%, which is ascribed to the weakly electron withdrawing effect of the chlorine atom. The reaction with N,N-dimethylaniline **1c** led to a mixture of

4-isomer (**2c**) and 2,4-isomer (**2c′**) in the presence of an electron-donating dimethylamino group. 1,3-dimethoxybenzene **1d** was only converted into 2,4-isomer **2d** due to the steric hindrance of the methoxy groups, giving a high 95% isolated yield. 1,2,3-trimethoxybenzene **1e** underwent bromination that gave 4-isomer **2e** as the only product. In the case of 1,3,5-trimethoxybenzene **1 f**, almost the same amount of 4-isomer **2f** and 2,4-brominated product **2f′** were obtained, giving a total yield of 81%. Using a mixture of $H_2O_2$ and HBr in dark, comparable yields of the brominated products have been obtained for the same substrates (Supplementary Table 7), which implies similar mechanisms (see below). Both PYT and PHT enable photocatalytic bromination of 1,3,5-trimethoxybenzene, when KBr was used as bromide source without addition of acid (Supplementary Table 8)[2]. Using KBr as bromide source is beneficial for bromination of electron rich acidophobic compounds, such as 3,4-ethylenedioxythiophene **1g**, which gave 2,5-dibromo-3,4-ethylenedioxythiophene **2 g** in 44% yield (Supplementary Table 9). In summary, bromination worked well with the substrates having strong electron-donating groups (EDG), such as –OMe and –$NMe_2$. In the presence of weak electron-withdrawing group (–Cl group), the reaction proceeded more slowly. Besides, the steric effect of the substituents is also an important factor in the reaction selectivity of aromatic hydrocarbons.

PYT was used as a model for unveiling the mechanism of the photocatalytic bromination of anisole. As evidenced by EPR spectroscopy (Fig. 5a), addition of anisole to PYT led to the increase of the specific number of free radicals in the material, from $2.69 \times 10^{17} g^{-1}$ to $2.81 \times 10^{17} g^{-1}$, upon irradiation. Therefore, in the studied reaction, anisole serves as a hole scavenger, which boosts the charge separation by electron transfer to the CTF.

It is well accepted that superoxide radical ($O_2^{\bullet-}$) and singlet oxygen ($^1O_2$) are two main reactive oxygen species that play a decisive role in photocatalytic reactions[61,62]. Given that the potential of PYT is located at −1.75 V vs. SCE (Supplementary Fig. 30), reduction of $O_2$ to $O_2^{\bullet-}$ ($O_2/O_2^{\bullet-}=−0.89$ V vs. SCE)[62]

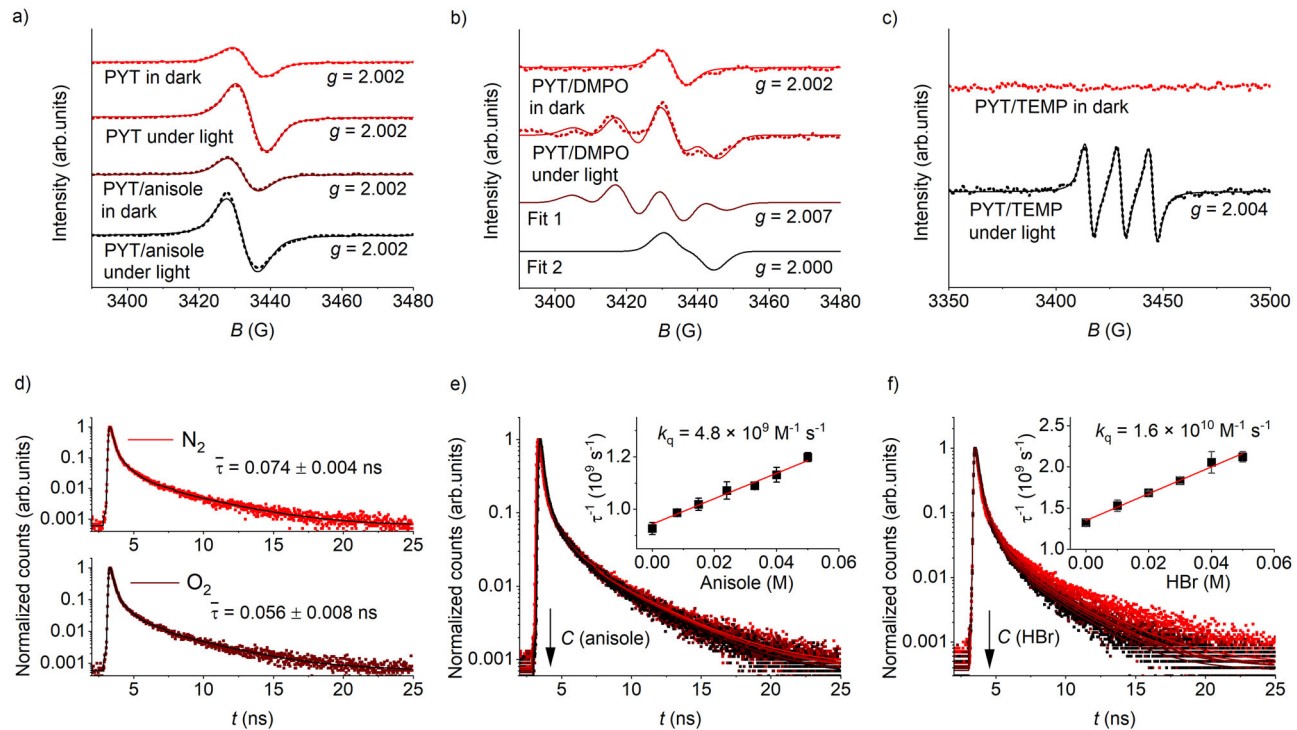

**Fig. 5 Mechanism study. a** EPR spectra of PYT and PYT/anisole. Dashed and solid curves show the raw and fitted data, respectively. The *g*-factor of each EPR spectrum is given in the graph. **b** PYT/DMPO. Fit 1 and fit 2 are the curves deconvoluted from the fitted curve in Fig. 4b. **c** PYT/TEMP. **d** TR-PL spectra of PYT dispersion collected at $\lambda_{exc} = 470$ nm and $\lambda_{em} = 580$ nm under $N_2$ and $O_2$. Mean ± std. dev. ($n = 3$). **e** TR-PL spectra of PYT dispersion collected at $\lambda_{exc} = 470$ nm and $\lambda_{em} = 580$ nm and Stern–Volmer plot (inset) in the presence of anisole under $N_2$. Mean ± std. dev. ($n = 3$). **f** The same as in Fig. 4e, but in the presence of HBr instead of anisole.

is thermodynamically favorable. To probe these two species, EPR spectra of PYT in the dark and under light irradiation were monitored, with addition of 5,5-dimethyl-1-pyrroline N-oxide (DMPO) and 2,2,6,6-tetramethylpiperidine (TEMP) as $O_2^{\bullet-}$ and $^1O_2$ scavengers, respectively. As shown in Fig. 5b, $O_2^{\bullet-}$ trapping upon irradiation gave an EPR spectrum comprising asymmetric signals, which derive from the overlapped signals of DMPO-$O_2$H adduct (fit 1, hyperfine coupling constant 12.5 G) and radicals produced by PYT (fit 2)[63]. Since $^1O_2$ was found to be hard to monitor, we decreased the PYT concentration in the PYT/TEMP mixture (to eliminate the interference of radicals in PYT), and increased both the irradiation power and time to obtain a clear spectrum. Indeed, the appearance of the TEMPO signal confirmed the formation of $^1O_2$ (Fig. 5c). To determine the contribution of $O_2^{\bullet-}$ and $^1O_2$ in the photocatalytic reaction, we carried out the reaction in the presence of benzoquinone ($O_2^{\bullet-}$ scavenger) and sodium azide (NaN$_3$) ($^1O_2$ scavenger). As a result, the product yield after 4 h was reduced from original 82% to 46% and 68%, respectively. This observation confirms the participation of both oxygen species in the photocatalytic process, with $O_2^{\bullet-}$ playing a more dominant role than $^1O_2$.

To further elucidate the reaction dynamics of the excited PYT, the emission decay profiles of PYT dispersion in MeCN/H$_2$O (volume ratio 5:1) were recorded with $\lambda_{exc} = 470$ nm (photon energy close to that used in the photocatalytic experiments) and $\lambda_{em} = 580$ nm by picosecond TR-PL spectroscopy. Figure 5d shows that the PL lifetime (amplitude average lifetime) of PYT dispersion is reduced from 74 to 56 ps under O$_2$. The oxygen-induced quenching is ascribed to the reaction of electrons with adsorbed O$_2$[64]. Effective quenching was also observed in the presence of anisole (Fig. 5e) and HBr (Fig. 5f), respectively. These data confirm the electron transfer from both the reagents to the photoexcited PYT[65], therefore implying a bromination via

electrophilic and nucleophilic substitution pathways. The corresponding quenching constants ($k_q$) determined from the Stern-Volmer plot (inset in Fig. 4e and f) are $4.8 \times 10^9$ and $1.6 \times 10^{10}$ M$^{-1}$ s$^{-1}$ in the case of anisole and HBr, respectively. Note that an intensity average lifetime was used for the calculation of $k_q$[53] (see the calculation method in the Supporting Information). The $k_q$ values are close to the diffusion limit ($\sim10^{10}$ M$^{-1}$ s$^{-1}$)[66], and indicate a likely faster electron transfer between excited PYT and HBr compared with anisole, presumably due to coupling with the transfer of a proton (proton-coupled electron transfer, PCET).

A reaction mechanism involving two possible (electrophilic and nucleophilic) pathways based on the above observations is shown in Supplementary Fig. 31 and discussed in Supplementary Note 4. Survey of conditions typically employed in photocatalytic oxidative halogenation clearly indicates that the presence of protons in the reaction mixture is mandatory[67]. On the other hand, protonation of semiconductors, such as carbon nitrides[68] and ZnO[69], shifts the VB potential to more positive values. In the context of oxidative bromination by CTFs, HBr not only provides protons for reduction of O$_2$ to HO$_2^{\bullet}$ or H$_2$O$_2$ via PCET (Supplementary Table 10), it protonates CTFs surfaces and as a result enables oxidation of more stable substrates (Supplementary Note 5). Indeed, both PYT and PHT enable oxidative bromination of anisole in acidic environment, but not in neutral (Supplementary Table 11).

**CTFs in dual Ni-photocatalytic C−N cross-coupling.** C–C and C–heteroatom cross-coupling of arylhalides with secondary amines[2,70], alcohols[71–73], thiols[74,75], trifluoroborates[76], and alkenes[77], respectively, mediated by a combination of cheap Ni salts and heterogeneous photocatalysts free of platinum group metals offers a convenient and scalable approach for the synthesis of value-added organic compounds. We have chosen pyrrolidine

**Table 2 Dual Ni-photocatalytic C-N coupling[a].**

Photocatalyst
NiBr$_2$·glyme
DABCO
Light
DMA, Ar, 48 h

**3a** + pyrrolidine → **4a** + **5a**

| Entry | Photocatalyst | Light | Time, h | 4a (%) | 5a (%) | Conversion (%) |
|-------|---------------|-------|---------|--------|--------|----------------|
| 1 | PYT | 400 nm | 48 | 39 | 0 | 39 |
| 2 | PHT | 400 nm | 48 | 100 | 0 | 100 |
| 3 | PYT | 465 nm | 48 | 5 | 0 | 5 |
| 4 | PHT | 465 nm | 48 | 71 | 29 | 100 |
| 5 | PYTnc | 465 nm | 48 | 70 | 30 | 100 |
| 6[b] | PHT | 465 nm | 48 | 100 | 0 | 100 |
| 7[c] | PHT | 465 nm | 48 | 44 | 0 | 44 |
| 8[d] | – | 465 nm | 48 | 0 | 0 | 0 |
| 9 | PHT | 525 nm | 120 | 24 | 76 | 100 |
| 10 | PYT | 525 nm | 120 | 5 | 0 | 5 |
| 11 | PHT | 625 nm | 120 | 0 | 68 | 70 |
| 12 | PYT | 625 nm | 120 | 0 | 0 | 0 |
| 13 | PYTnc | 625 nm | 168 | 0 | 7 | 7 |
| 14 | PHT | 625 nm | 168 | 0 | 89 | 91 |
| 15[e] | PHT | 625 nm | 168 | 0 | 80 | 80 |

[a]Reaction conditions: photocatalyst (12 mg), 4-bromobenzonitrile **3a** (9.1 mg, 0.05 mmol), pyrrolidine (7.4 μL, 0.09 mmol), NiBr$_2$·glyme (0.8 mg, 0.0025 mmol), DABCO (12.3 mg, 0.11 mmol), N,N-dimethylacetamide (1 mL), 48 h. Condition was adapted from Ref. [2]. Yield and conversion were determined by GC-MS.
[b]1,10-phenanthroline (0.0025 mmol) was added as a ligand.
[c]Without NiBr$_2$·glyme.
[d]Without photocatalyst.
[e]Data obtained after using PHT for three consecutive rounds.

and 4-bromobenzonitrile **3a** as reaction partners, NiBr$_2$·glyme as precatalyst, and DABCO as base[2]. Upon illumination of the reaction mixture with 400 nm photons, the dehalogenation product **4a** was obtained selectively with 39% and 100% yield in case of PYT and PHT, respectively (Table 2, entries 1,2). Despite extended absorption, upon illumination with 465 nm photons PYT gave **4a** with 5% yield, but failed to produce **5a** (entry 3). PHT and PYTnc gave **5a** with 29% and 30%, respectively, but the main product was **4a** (entry 4,5). Addition of phenanthroline as a ligand to the system containing PHT led to selective formation of **4a** (entry 6). Without any Ni catalyst, PHT gave dehalogenation product with 44% yield (entry 7), while the reaction did not proceed without any photocatalyst (entry 8). Similar to oxidative halogenation of anisole with PYT, a control experiment to check for "leaching" of organics from PHT that could sensitize the dual-Ni photocatalytic C−N cross-coupling was negative – neither **5a** was formed nor any conversion of **3a** took place. Under illumination with 525 nm photons for 120 h, PHT gave **5a** with 76% yield (entry 9), while PYT – only product **4a** with 5% yield (entry 10). To suppress completely the formation of the dehalogenation product **4a**, we switched to red (625 nm) photons. PHT gave **5a** with 68% yield (entry 11). PYT did not convert any of the reagents (entry 12), while PYTnc gave **5a** with 7% (entry 13). The yield of **5a** was further improved to 89% without compromising the selectivity by extending the reaction time under red light to 168 h (entry 14). Longer reaction times are quite common, when photons of longer wavelengths are employed in the dual Ni-photocatalysis[70]. Optimizing, for example, the relative distance between the sensitizer and the transition metal site emerged as a viable strategy to improve the quantum efficiency of the reaction. This has been demonstrated for an Ir-sensitizer and a Ni-catalyst integrated into a soft polymer[78]. After three rounds of use, PHT gave **5a** with 80% yield (entry 15). A set of

techniques unambiguously confirmed that the chemical structure of PHT remained intact throughout the dual Ni-photocatalysis, which is explained by extremely mild conditions – low-energy electromagnetic radiation (625 nm) and 25 °C (Supplementary Note 6).

Overall, PHT enables selective coupling of electron-deficient arylhalides **3a-d** with 18%–89% yields (Fig. 6), while electron-rich substrates **3e-g** failed to give the coupling products (Supplementary Fig. 32), which is one of the existing challenges in transition-metal catalyzed cross-coupling reactions[79,80].

## Discussion

The above results clearly speak for the efficacy of our designed strategy. Here, we discuss our understanding of how the molecular structural control of CTFs translates into new properties and ultimately influences their photocatalytic activity. A photocatalytic reaction is initiated by photon absorption by a photocatalyst to generate electron-hole pairs followed by their separation to drive redox reactions. As observed earlier[11], the substitution of the C−H moiety with nitrogen atoms can induce an increase in planarity of the platform, thus leading to a higher degree of conjugation and hence a redshift of the absorption. Indeed, nitrogen rich PYT possesses: i) a narrower optical gap of 2.50 eV compared with 2.72 eV for PHT and ii) extended absorption in the nIR (Fig. 3a).

Secondly, charge carriers can undergo rapid recombination before they can dissociate at the polymer/solution interface. The construction of proper electron donor–acceptor systems, in our case, a system with larger acceptor domains, can be expected to promote the charge separation, thereby minimizing the exciton loss before it decays back to the electronic ground state. As evidenced by TR-PL data (Fig. 3e, Supplementary Fig. 33) excited

state dynamics in PHT, PYT, and PYTnc is different and depends strongly on the excitation wavelength. As deduced from Stern-Volmer plots shown in Fig. 5e, f, electron transfer between PYT and anisole occurs at the rate of $4.8 \cdot 10^9$ M$^{-1}$ s$^{-1}$; between PYT and HBr – $1.6 \cdot 10^{10}$ M$^{-1}$ s$^{-1}$, which is limited by the diffusion of the reagents to the surface of the photocatalyst. Therefore, excitons that are separated with a rate smaller (or time greater) than the rate of electron transfer do not enable redox reaction. Figure 7a shows the correlation between the yield of 4-bromoanisole and *the longest time* ($\bar{\tau}_{max}$) of photogenerated charge carriers separation upon excitation at 470 and 640 nm. Therefore, the faster separation of the photogenerated charge carriers the higher is the yield of 4-bromoanisole under illumination with light of the corresponding wavelength. Despite the higher activity of photocatalysts (higher yield of a product) in certain reactions is

typically explained by improved absorption of photons of certain wavelength, our results indicate that absorption itself may not be the only factor. Despite PHT has significantly lower absorption at 620–625 nm compared with PYT (Fig. 3a), it produces 4-bromoanisole with a higher rate, that is, 27% yield versus 12% (Table 1).

Potentials of the occupied states, VB or intraband states laying close to the VB, define the driving force for electron transfer from the substrate. Therefore, selective excitation of either n-π* or π-π* transitions in the conjugated polymers, such as carbon nitrides, can alter the selectivity of a chemical reaction[21,22]. Analyses of TCSPC data and steady-state emission spectra in combination with the results of CTFs photocatalytic activity in oxidative bromination and dual Ni-photocatalytic C-N cross-coupling suggest that the concept is applicable for our systems as well. Thus, excitation of CTFs with shorted wavelength, such as λ ≤ 450 nm for PHT, λ ≤ 500 nm for PYT and λ ≤ 650 nm for PYTnc as deduced from Tauc plots (Supplementary Fig. 7b), gives the excited state with π-π* character. On the other hand, excitation of CTFs with longer wavelength, such as λ > 450 nm for PHT, λ > 500 nm for PYT and λ > 650 nm for PYTnc, give excited state with n-π* character. These excited states possess oxidation power that is defined by the band structure and the energy of incident photons (Relation 1).

Whenever a reaction occurs at the interface (e.g., heterogeneous catalysis), porosity may play a critical role. Some studies did not show a direct correlation between the degree of porosity and activity[16], while others reported a marked effect[81]. In the context of oxidative bromination, an increase in surface area and pore volume in PYT may favor photocatalytic reaction by providing a larger number of basic sites for hosting protons, which facilitate oxidation of the substrates via multisite PCET – protons are supplied by HBr, while electrons by the substrate. Larger surface area and higher number of basic sites in PYT compared with PHT, facilitates trapping of electrons under ambient conditions as deduced from EPR spectra (Fig. 3f). In the photocatalytic cycle, compensation of the excessive negative charge on the surface of the photocatalyst by coupling it with H$^+$ stabilizes the system by ΔG$_{st}$, which for PYT is as high as 1.06 eV when taken into account the oxidation potentials of the substrates from Fig. 3 and VB potential of CTFs (Fig. 7b, Supplementary Fig. 30). As a result, energy of photons at the PET step is directed toward the oxidation of thermodynamically more stable substrates.

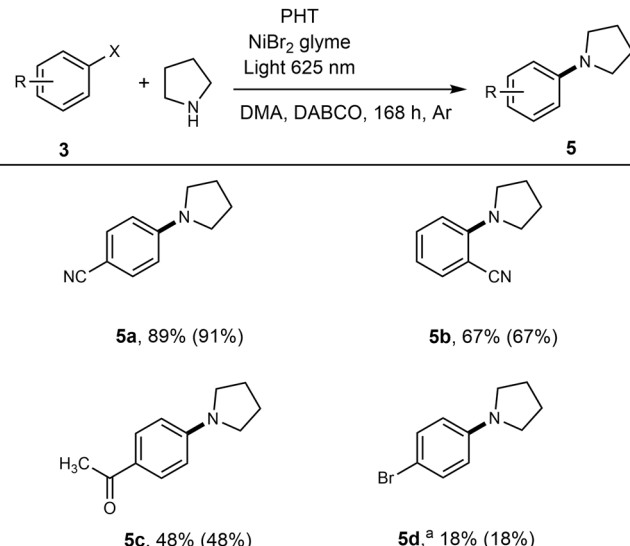

**Fig. 6 Scope of dual Ni-photocatalytic C–N cross coupling.** Arylhalide (0.05 mmol), PHT (12 mg), pyrrolidine (7.4 μL, 0.09 mmol), NiBr$_2$·glyme (0.8 mg, 0.0025 mmol), DABCO (12.3 mg, 0.11 mmol), N,N-dimethylacetamide (1 mL), Light 302 mW cm$^{-2}$, 168 h. Yield and conversion (in parentheses) determined via GC-MS. $^a$From 1-Bromo-4-iodobenzene.

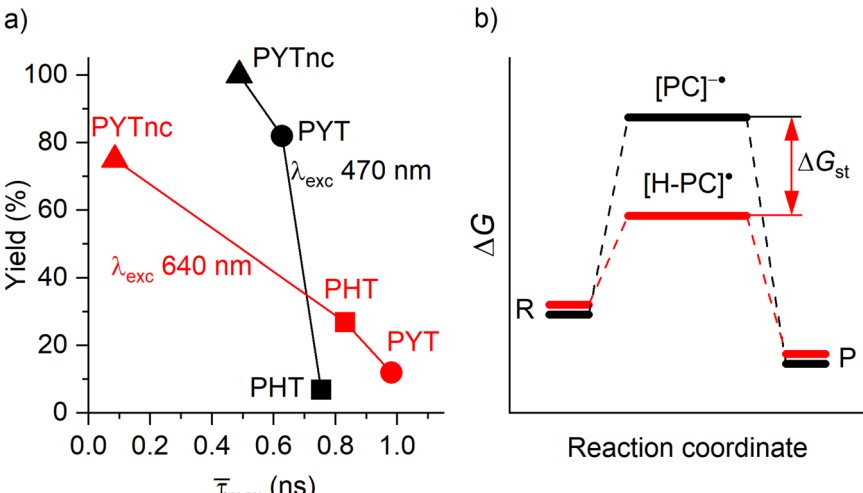

**Fig. 7 Discussion of the structure-activity relationship. a** Correlation of the yield of 4-bromoanisole **2a** with $\bar{\tau}_{max}$; **b** Stabilization of the photocatalyst excited state by protonation of its surface allows utilizing energy at the PET step to oxidize thermodynamically more stable substrates. "R" stands for reagents, "P" stands for products.

However, porosity is neither the only nor the primary factor that determines the CTF performance. If this would be the case, due to, for example, improved accessibility of the substrates, PYT, which features a four times larger surface area and six times larger cumulative pore volume than PHT, would be more active in all of the photocatalytic experiments. Moreover, PYT would be more active than PHT regardless of the photon wavelength, which was selected for the photocatalytic experiments. This is, however, not the case (Supplementary Fig. 34). In fact, the CTF activity correlates with its surface area and porosity only in the oxidative halogenation of anisole under 468 nm illumination. But, PHT, which has a lower surface area as well as lower porosity, is more active in oxidative halogenation under 625 nm illumination and dual Ni-photocatalytic C–N coupling under 465, 525, and 625 nm illumination.

In the case of C–N cross-coupling, analysis shown in Fig. 6a did not give a trend likely because the reaction is mediated by Ni-catalyst, while CTF in combination with light facilitate destabilization of Ni-intermediates postulated in the catalytic cycle either via electron or energy transfer[82,83]. In particular, reductive elimination from Ni(II)-intermediate is endothermic[84,85], but becomes exothermic if Ni is in oxidation state III (Supplementary Fig. 35). Several conclusions related to the band structure of CTFs and their activity could be made taking into account mechanism of dual Ni-photocatalytic C–N cross-coupling suggested in the literature, which is based on one-electron transfer[86]. Overall, relative alignment of band edges in PHT, which upon photon absorption gives the excited state that is more oxidative and less reductive compared with PYT, is beneficial to generate tentative Ni(I) and Ni(III) species (Supplementary Note 7). On the other hand, as inferred from transient absorption spectroscopy, the lifetime of the excited states exceeds >50 µs for all CTFs studied herein (Supplementary Note 3). Therefore, energy transfer pathway cannot be excluded[87,88].

Earlier, Pieber et al. have shown that wavelength of incident photons is one of the leverages to decrease the rate of reductive elimination in the catalytic cycle and therefore mitigate deactivation of Ni-catalyst[70]. Our results, presented in Table 2, indicate that by decreasing the energy of incident photons, we effectively suppress undesirable dehalogenation process and improve selectivity toward the product of C–N cross-coupling.

We believe that the photocatalytic reaction is a complex process where multiple factors are at play. Our study suggests that not only the structure of amorphous carbon-based materials, but also the wavelength of incident photons strongly affect charge separation and redox properties. As such tuning the chemical structure of the material and selection of the light source with appropriate wavelength can be used to tune performance of the material in the photocatalytic reaction.

In conclusions, two amorphous CTFs with extended light absorption in the visible range were prepared, instrumentalizing the gap between the conduction and valence band edges with the intraband states that originate from nitrogen lone pairs embedded into the framework. In PHT, which possesses p-phenylene linkers, the intraband states are located in the vicinity of the VB, while in PYT possessing pyridine-2,6-diyl linkers they are in the middle of the band gap. The intraband states are weakly emissive and photocatalytically active. Two reactions, oxidative bromination of electron-rich aromatic compounds and dual Ni-photocatalytic C–N cross-coupling, are mediated by the prepared CTFs. In oxidative bromination, the intraband states allow the reaction to proceed under illumination with 625 nm photons. The yield of the products correlates with the time of photogenerated charge carriers separation. The role of acidic medium in oxidative halogenation has been explained by a stabilizing effect of $H^+$ on the reductively quenched photocatalyst when

coupled with the transfer of electron. Due to the relative alignment of band edges in PHT that upon excitation gives more oxidative and less reductive excited state compared with PYT, CTF mediates dual Ni-photocatalytic C–N cross-coupling of electron-deficient arylhalides and pyrrolidine. Using 625 nm photons, we suppressed completely the undesirable dehalogenation process and maximized the yield of the cross-coupling product. The mechanism of photocatalytic reactions mediated by CTFs is complex, but due to several features, such as relative alignment of band edges, the presence of intraband states, and an ensemble of basic pyridinic moieties, they offer several modes for substrate activation, which in homogeneous photocatalysis are typically achieved by combining several molecular mediators. We envision that progress in controlled synthetic approaches combined with careful structural and photophysical characterizations are promising avenues to the establishment of precise structure-property relationships in nanomaterials, which will undoubtedly increase reaction control and thereby pave the way to large-scale applications of polymer-based photocatalysts.

## Methods

**Synthesis of PHT and PYT.** The terephthalamidine dihydrochloride precursors were synthesized based on reported procedure[89]. Terephthalamidine dihydrochloride (1177.5 mg, 5 mmol) and either 2,6-pyridinedicarboxaldehyde (337.8 mg, 2.5 mmol) (for PYT) or isophthalaldehyde (335.3 mg, 2.5 mmol) for (PHT), and cesium carbonate (3580 mg, 11 mmol) were added to a solution of DMSO (25.0 mL) and water (1 mL) in 50 mL round-bottom flask. The mixture was heated at 100 °C for 24 h, then up to 150 °C for 36 h. After cool down to room temperature the resulting precipitate was washed with diluted HCl (3 × 20 mL) to remove the salt and residual cesium carbonate, and washed with water (3 × 30 mL), acetone (3 × 10 mL), and THF (3 × 10 mL), before drying at 80 °C under vacuum for 12 h. Finally, PYT and PHT were obtained as yellow powders.

**Photocatalytic oxidative bromination of anisole (screening reactions).** A glass vial (4 mL) was charged with a mixture of anisole (2.2 µL, 20 µmol), HBr (0.1 mL, 48 wt.%), photocatalyst (4 mg), and acetonitrile (0.5 mL). Magnetic stir bar was placed in the vial. The vial was purged with $O_2$ for 30 s and closed with cap. The reaction mixture was vigrously stirred in the photoreactor with blue LED module 1 (14 mW cm$^{-2}$) for 24 h. After the reaction, $CDCl_3$ (0.7 mL) and water (0.1 mL) were added to the reaction mixture. The organic layer was separated and measured with $^1H$ NMR using 1,4-dinitrobenzene as internal standard. The catalyst in aqueous phase was separated by centrifugation, washed with acetonitrile for three times, and dried for recycle use.

## Data availability

The datasets generated during and/or analyzed during the current study are available from the corresponding author on reasonable request.

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

## Acknowledgements

Y.Z. acknowledges China Scholarship Council (CSC) for providing a scholarship (CSC No. 201906280093). M.A. and A.S. gratefully acknowledge Max Planck Society for the financial support of this project. J.S. acknowledges the National Natural Science Foundation of China (21972110). P.V.D.V., P.G.D., and S.A. acknowledge the UGent concerted action grant BOFGOA2017000303, Ghent University BOF doctoral grant 01D04318 and the Research Foundation Flanders (FWO-Vlaanderen) grant no. G000117N. Authors acknowledge Miss Jiamei Liu at Instrument Analysis Center of Xi'an Jiaotong University for her assistance with XPS analysis. Authors acknowledge Katharina ten Brummelhuis for the low-temperature EPR spectra measurements; Dr. Nadezda V. Tarakina and Dr. Tobias Heil for TEM analysis.

## Author contributions

Y.Z. characterization of CTFs, screening of CTFs in oxidative halogenation, and manuscript writing; S.A. synthesis of CTFs, manuscript writing; P.G.D. synthesis of CTFs, manuscript writing; S.M. characterization of CTFs, dual Ni photocatalytic C–N cross coupling; C.M.S. TAS study; D.C. UPS study; P.V.D.V. manuscript writing; J.W.S. manuscript writing; M.A. manuscript writing; D.M.G. TAS study, manuscript writing; A.S. idea, coordination of the project, manuscript writing.

## Funding

## Competing interests

The authors declare no competing interests.
