## [Peer Review File · Nature Communications]

Title: Red Edge Effect and Chromoselective Photocatalysis with Amorphous Covalent Triazine-based FrameworksREVIEWER COMMENTS

Reviewer #1 (Remarks to the Author):

In this manuscript, Savateev et al. reported a molecular structural design of two-dimensional amorphous covalent triazine-based frameworks (CTFs), which possesses a strong red edge effect (REE).

Characterizations have demonstrated that the nitrogen substitution in the CTFs could modulate the porosity, optical gaps, separation of charge carriers etc. The obtained CTFs show enhanced visible-light driven bromination of aromatic compounds using HBr and O₂, even under low energetic red light. A detailed study was also performed to figure out the structure-activity relationship of the obtained CTFs as well as the photocatalytic mechanism. The manuscript is interesting, impressive and well-organized. Therefore, publication in Nature Communications is recommended after minor revision.

1. The authors point out that the presence of several redox sites in amorphous CTFs is based on strong REE effect. And in crystalline CTFs that do not show REE only one oxidation site close to the VBM participates in the electron transfer. It will be helpful if the PYT with improved crystalline structure that has weak REE should be synthesized as a control, to fully study the real-contribution of strong REE effect on its photocatalytic performance.
2. In the visible-light-driven bromination of aromatic compounds, could the HBr be replaced by NaBr? Does the proton play any role in such bromination reactions?
3. The reaction-scope of this protocol is quite narrow. Please check whether electron-rich heterocycles could be the suitable substrates or not.
4. Supplementary experiments to detect the generated H₂O₂ intermediates and the influence of additional introduced H₂O₂ oxidant on photocatalytic bromination should be provided.
5. Although the authors proposed an experimental assumption for the excitation profiles based on the experimental data (Fig. 2), the current investigation of band diagram in this work is not enough to support the excitation process. First of all, the characterization of VBM is missing in this work. Secondly, a Mott-Schottky measurement is not enough as well. Different frequencies should be done to confirm the CBM of the material.
6. In figure 3a, the insert text "PYT/anisole" should be changed to "PYT/anisole".
7. The format errors in the references sections should be carefully examined (i. e. ref 26).
8. Covalent organic frameworks (COFs) generally refer to the one with well-defined crystalline porous structures. Please carefully check that whether such kind of amorphous polymers could be defined as COFs.

Reviewer #2 (Remarks to the Author):

In this work Savateev et al. synthesize two new amorphous CTFs and they were used for photocatalysis, bromination of arenes. For the synthesis it is employed 2,6-pyridinedicarboxaldehyde (PYT) or an aryl aldehyde (PHT), demonstrating the importance of pyridinic nitrogen. Later, they carry out the characterization of the two materials, using the usual techniques for these materials, such as XPS, BET,

or IR. The materials have a very low porosity (104 and 27 m²g⁻¹), being amorphous, and limiting their use to other types of applications (H₂, CO₂, etc.). The optical properties obtained are very well performed, using different complementary techniques for the characterization of both materials, and showing that the incorporation of pyridine has a substantial impact on the photoredox properties. In the last part of the work, the authors study the bromination of aromatic rings, where the only advantage in the use of the material is the in situ generation of the electrophilic species HOBr. The material does not have any impact on the reactivity of the substrates because it just depends on the HOBr that is generated in the reaction media. Indeed, there is a final discussion of the structure activity of COF. The reactivity is occurring near the surface of the material or even in the reaction media. Therefore, the only impact of the material is the amount of the HOBr that the CTF is able to produce.

In conclusion, although the work is technically very well performed, the reactivity explored with this material is not very new, nor is a very important advantage observed in the use of this material. In my opinion, it should be published in a more specific material journal.

Point-by-point replies to the referees' comments to the article "Red Edge Effect and Chromoselective Photocatalysis with Amorphous Covalent Triazine-based Frameworks" (NCOMMS-21-07813).

Referees comments in bold black

Our replies in blue

Data and discussion added to the manuscript and SI are highlighted with orange background

REVIEWER COMMENTS

Reviewer #1 (Remarks to the Author):

In this manuscript, Savateev etc. reported a molecular structural design of two-dimensional amorphous covalent triazine-based frameworks (CTFs), which possesses a strong red edge effect (REE). Characterizations have demonstrated that the nitrogen substitution in the CTFs could modulate the porosity, optical gaps, separation of charge carriers etc. The obtained CTFs show enhanced visible-light driven bromination of aromatic compounds using HBr and O₂, even under low energetic red light. A detailed study was also performed to figure out the structure-activity relationship of the obtained CTFs as well as the photocatalytic mechanism. The manuscript is interesting, impressive and well-organized. Therefore, publication in Nature Communications is recommended after minor revision.

Response: We appreciate referee's time dedicated to evaluate our work, finding our manuscript interesting and recommendation for publication in Nature Communications. We also appreciate constructive comments that we addressed as indicated in point-by-point responses below.

1. The authors point out that the presence of several redox sites in amorphous CTFs is based on strong REE effect. And in crystalline CTFs that do not show REE only one oxidation site close to the VBM participates in the electron transfer. It will be helpful if the PYT with improved crystalline structure that has weak REE should be synthesized as a control, to fully study the real-contribution of strong REE effect on its photocatalytic performance.

Response: Indeed, it is an intriguing question whether crystallinity of the material correlates with the REE. To address this question,

a more crystalline material possessing pyridine-2,6-diyl linkers (PYTnc) was synthesized by polycondensation of 2,6-bis[(phenylimino)methyl]pyridine with terephthalamidine dihydrochloride (Supplementary Figure 10).

Supplementary Figure 10. Synthesis and ideal structure of PYTnc.

Using the Schiff base instead of pyridine-2,6-dicarboxaldehyde, polycondensation is expected to proceed slower and as a result would yield more ordered material. Improved crystallinity of PYTnc compared to PYT is supported by more pronounced diffraction peak at 18.05 degrees in the PXRD (Supplementary Figure 2).

Supplementary Figure 2. Powder XRD patterns of PHT, PYT and PYTnc.

Still, PYTnc is nanocrystalline material with the crystallites diameter estimated from the Scherrer equation to be ca. 3 nm.

$$\tau = \frac{K \cdot \lambda}{\beta \cdot \cos\theta}$$

where τ – the mean size of the crystallite, nm; K – dimensionless shape factor, assumed to be 0.9; λ – wavelength of the diffractometer, nm; β – broadening (FWHM) of the diffraction line, degree; θ – Bragg angle, degree.

PYTc was characterized by a set of spectroscopic techniques, similar to PYT and PHT. In the DRUV-vis absorption spectrum (Supplementary Figure 7a), PYTnc shows absorption edge at ca. 600 nm. Using this absorption edge, the optical gap defined by π - π^* transitions was calculated to be 1.92 eV (Supplementary Figure 7b). As indicated by the absorption features stretching up to nIR, PYTnc also contains intraband states.

Supplementary Figure 7. Spectroscopic characterization of materials. a) DRUV-vis absorption spectrum of PYTnc. Some discontinuous data points caused by the change in detector at ~ 900 nm were removed. b) Tauc plots of PHT, PYT and PYTnc; π - π^* transitions were taken as primary to determine optical band gap (E_g) in the materials.

Upon excitation at λ_{exc} 350 nm, PYTnc emits photons with the maximum at 650 nm (Supplementary Figure 34).

Supplementary Figure 34. PL spectra of PYTnc acquired at λ_{exc} 350, 365, 400, 450, 500, 550, 600 and 650 nm.

Compared to PYT, the emission peak is red-shifted due to narrower optical gap. Similar to PHT and PYT, Stokes shift in PYTnc is ca. 50 nm at λ_{exc} 350 nm. When λ_{exc} exceeds ca. 550 nm, the emission peaks starts progressively shifting to longer wavelength – PYTnc also shows REE. The optical gap defined by π - π^* transitions correlates with the onset of REE in the studied materials (Supplementary Figure 16a,b).

Supplementary Figure 16. Correlation between REE and E_g in COFs. a) Plot of emission maximum (λ_{em}) versus excitation wavelength (λ_{exc}). Note two regions for each curve: no REE (position of the emission maximum peak remains nearly constant regardless the λ_{exc}) and REE

(position of the emission maximum peak follows the λ_{exc}). b) Correlation between REE onset and optical gap defined by $\pi-\pi^*$ transitions. REE onset was determined as average of two adjacent λ_{exc} values in panel (a) at which REE and no REE was observed followed by conversion into eV using Planck-Einstein relation.

Unlike to PHT and PYT, in TR-PL spectra, PYTnc shows non-monotonous dependence of the amplitude average lifetime ($\bar{\tau}$, Supplementary Figure 32). Thus, upon excitation at 375 nm the maximum amplitude average lifetime is 0.330 ns, followed by 0.490 ns (upon excitation at 470 nm) and 0.087 ns (upon excitation at 640 nm).

Supplementary Figure 32. Amplitude average lifetime of PYTnc obtained with $\lambda_{exc} = 375$, 470 and 640 nm and $\lambda_{em} = 400-900$ nm.

2. In the visible-light-driven bromination of aromatic compounds, could the HBr be replaced by NaBr? Does the proton play any role in such bromination reactions?

Response: To test if bromination can proceed under neutral pH using alkali metal salt, we adopted a procedure from Science 2019, 365, 360-366. Anisole did not react under such conditions. Bromination of anisole using KBr was not reported in the abovementioned article either.

Supplementary Table 9. An attempt to enable photocatalytic oxidative bromination of anisole with PYT using KBr as bromine source.^a

Entry	Photocatalyst	Light	Yield (%)	Conversion (%)
1	PYT 12 mg	400 nm	0	0
2	PHT 12 mg	400 nm	0	0
3	PYT 12 mg	465 nm	0	0
4	PHT 12 mg	465 nm	0	0

^a Reaction conditions: anisole (21.7 μ L, 0.2 mmol), KBr (119 mg, 1 mmol), 48 h, DMSO-*d*₆:H₂O (1 mL, 9:1), O₂ (1 bar). Condition were adopted from the reference ¹. Yield and conversion were determined by ¹H NMR with internal standard.

On the other hand, 1,3,5-trimethoxyanisole gave the bromination product, which was also reported in the reference article.

Supplementary Table 6. Oxidative photocatalytic bromination of 1,3,5-trimethoxybenzene with KBr.^a

Entry ^a	Catalyst	Light	Yield (%) ^b	Conversion (%) ^b
1	PYT 12 mg	400 nm	85	100
2	PHT 12 mg	400 nm	83	100
3	PYT 12 mg	465 nm	52	88
4	PHT 12 mg	465 nm	14	14

^a Reaction conditions: anisole (21.7 μ L, 0.2 mmol), KBr (119 mg, 1 mmol), 48 h, DMSO-*d*₆:H₂O (1 mL, 9:1), O₂ (1 bar). Condition were adopted from the reference ¹. Yield and conversion were determined by ¹H NMR with internal standard.

Such observation we explain by higher stability of anisole against oxidation ($E = +1.81 \text{ V vs SCE}$)² compared to 1,3,5-trimethoxybenzene ($E = +1.49 \text{ V vs SCE}$).³

Supplementary Figure 29. Potentials of VB and CB in PHT and PYT and redox potentials of organic molecules. Oxidation potentials of aniline,² 1,2,3-trimethoxybenzene,³ 1,3,5-trimethoxybenzene,³ 1,3-dimethoxybenzene² and anisole² and reduction potential of O_2 to $\text{O}_2^{\bullet -}$ (in acetonitrile)⁴ in electrochemical scale (vs SCE). Potentials of VB and CB in PHT and PYT were calculated by converting CB and VB energy levels (eV, shown in Figure 3b) using equation: $U = -k \cdot (E - U_{\text{AVS/SHE}}) - U_{\text{SCE/SHE}}$, where E – CB or VB energy level, eV; k – conversion factor, $1 \text{ V} \cdot \text{eV}^{-1}$; $U_{\text{AVS/SHE}}$ – potential of standard hydrogen electrode expressed in physical scale, $+4.44 \text{ eV}$; $U_{\text{SCE/SHE}}$ – potential of SCE versus SHE, $+0.244 \text{ V}$.⁵

In general, presence of proton in the reaction mixture is mandatory for oxidative halogenation of organic compounds.⁶ The role of acid in photocatalytic oxidative halogenation of anisole is summarized in Supplementary Table 8.

Supplementary Table 8. Role of acid in photocatalytic oxidative halogenation of electron rich aromatic compounds.

Entry	Photocatalyst	Role of acid	Reference
1	Microporous organic polymers (heterogeneous)	Participates in PCET to generate HO_2^{\bullet} from O_2 .	7
2		Protonation of the photocatalyst increases excited oxidation potential of the photocatalyst, which in turn enables	8

	sodium anthraquinone-2-sulfonate (SAS, homogeneous)	oxidation of thermodynamically more stable substrates via PCET. SAS-H• as the intermediate in the photocatalytic cycle is postulated.	
3	 Riboflavin tetraacetate (RFT) (homogeneous)	Acetic acid is converted into peracetic acid in situ, which in turn enables oxidation of Cl ⁻ to OCl ⁻ .	9
4	Potassium poly(heptazine imide)	Participates in PCET to generate H ₂ O ₂ from O ₂ .	10
5	 [Acr-Mes] ⁺ ClO ₄ ⁻	Participates in PCET to generate HO ₂ [•] from O ₂ .	11

Analysis of our own results and results obtained by other research groups, point that in case of nitrogen-rich conjugated organic polymers, the role of proton in the photocatalytic mechanism is not only to participate in reduction of O₂ to H₂O₂ (or HO₂[•]) via PCET, protons in fact shift VB level to more positive values. Such conclusion is in agreement with the results obtained by Wu et al. for protonated g-C₃N₄.¹² Therein, protonation shifts the VB potential by 0.77V from 1.6 V to 2.37 V vs NHE.¹² Stabilization by 0.77 V also pulls down the potential of the CB, so that optical gap remains nearly constant. Protons (and in general cations) shift the CB and VB levels in nanoparticles of inorganic semiconductors, such as ZnO, to more positive values.¹³ Protonation of a photocatalyst surface increases oxidation power of the photogenerated holes compared to that under neutral pH.

Using structure of PYT as an example, schematically the mechanism of oxidative bromination of anisole is depicted in Supplementary Figure 38.

Supplementary Figure 38. Schematic mechanism of anisole oxidative bromination with PYT. In this figure only electron transfer is considered. Tentative local structure of PYT intermediates is shown. Upon excitation with light, electron is transferred from electron enriched part (*p*-phenylene units) to electron-deficient part (pyridine-2,6-diyl and/or triazine-linkers) of PYT.

Protonation of certain pyridine units in PYT gives $[H-PYT]^+ Br^-$. Upon excitation, electron transfer from the VB, which is localized at *p*-phenylene moieties (the electron-enriched part of the polymer) to the CB, which is localized at protonated pyridine-2,6-diyl and/or triazine-linkers (the electron-deficient part of the polymer), gives excited state $[H-PYT]^{+*} Br^-$. One-electron oxidation of anisole followed by nucleophilic attack of bromide anion and abstraction of hydrogen atom by hydroperoxyl radical gives 4-bromoanisole and transient $[H-PYT]^*$. Open-shell character of the tentative $[H-PYT]^*$ intermediate has been partially confirmed by increase of the amplitude of the EPR signal upon illumination of the mixture of PYT and anisole (Figure 4a). The latter transfers hydrogen atom to oxygen, which closes the photocatalytic cycle.

The suggested mechanism can rationalize the fact that in acidic environment we obtained 4-bromoanisole from anisole, despite its oxidation potential +1.81 V vs SCE² is more positive than the VB potentials of PHT (+1.55 V vs SCE) and PYT (+0.75 V vs SCE). Therefore, in acidic environment the VB potential of PHT is shifted by at least 0.26 V to more positive values. As a result, oxidation of anisole to the photocatalyst becomes feasible. Taking into account the band diagram of PYT, such shift is even larger, ca. 1.06 V, but due to higher content of pyridinic-nitrogen atoms (basic sites where protons reside) and 4 times larger surface area, is apparently plausible.

Overall, acidic conditions allow expanding the scope of the substrates in the oxidative halogenation as long as functional groups are tolerated.

3. The reaction-scope of this protocol is quite narrow. Please check whether electron-rich heterocycles could be the suitable substrates or not.

Response: Using PYT and KBr at neutral pH, we performed bromination of EDOT, which is a precursor for synthesis of conductive polymers. Dibromo derivative was obtained with 44% yield upon illumination the reaction mixture at 465 nm.

Supplementary Table 7. Photocatalytic oxidative bromination of 3,4-ethylenedioxythiophene with PYT using KBr as bromine source.

Entry ^a	Light	Yield (%) ^b	Conversion (%) ^b
1	465 nm	44	100
2	400 nm	42	100

^a Reaction conditions: PYT (12 mg), 3,4-ethylenedioxythiophene (21.4 μL, 0.2 mmol), KBr (119 mg, 1 mmol), 48 h, DMSO-*d*₆:H₂O (1 mL, 9:1), O₂ (1 bar). Condition were adopted from the reference ¹. Yield and conversion were determined by ¹H NMR with internal standard.

We also tested benzofuran and pyrrole derivatives under neutral pH using KBr as bromide source. However, a complex mixture was obtained in this case.

4. Supplementary experiments to detect the generated H₂O₂ intermediates and the influence of additional introduced H₂O₂ oxidant on photocatalytic bromination should be provided.

Response:

Addition of variable quantities of H₂O₂ into the reaction mixture gave 4-bromoanisole with lower yield, which may be explained by overoxidation of the product under such conditions (Supplementary Table 4). These results obviously point at advantages of photocatalytic generation of H₂O₂ in situ.

Supplementary Table 4. Influence of explicitly added quantities of H₂O₂ on the yield of 4-bromoanisole.

Entry	H ₂ O ₂	Yield (%)	Conversion (%)
1	0.6 μL (0.01 eq.)	85	100
2	3 μL (0.05 eq.)	89	100
3	12 μL (0.2 eq.)	79	100
4	30 μL (0.5 eq.)	72	100
5	60 μL (1 eq.)	75	100
6	120 μL (2 eq.)	77	100

Reaction conditions: anisole (65 μL, 0.6 mmol), PYT (4 mg), HBr (0.6 mL, 6 mmol, 48 wt. %), 468 nm (101 mW cm⁻²), 48 h, MeCN (3 mL), O₂ (1 bar).

We confirmed formation of H_2O_2 by EPR spectroscopy using DMPO as a radical trap (Figure 4b). In such experiment a signal of DMPO- O_2H adduct was detected.

Figure 4. Mechanism study. b) PYT/DMPO. Fit 1 and fit 2 are the curves deconvoluted from the fitted curve in 4b.

5. Although the authors proposed an experimental assumption for the excitation profiles based on the experimental data (Fig.2), the current investigation of band diagram in this work is not enough to support the excitation process. First of all, the characterization of VBM is missing in this work. Secondly, a Mott-Schottky measurement is not enough as well. Different frequencies should be done to confirm the CBM of the material.

Response: As suggested by the referee, we determined VBM in the PHT and PYT using ultraviolet photoelectron spectroscopy (UPS).

The VB levels in COFs were determined from photoelectron spectra (Supplementary Figure 8). The CB levels in COFs were calculated by adding E_g values derived from Tauc plots to the VB levels and are -3.51 and -2.93 eV for PHT and PYT respectively (Figure 2b). From Mott-Schottky plots (Supplementary Figure 9), flat band potentials (E_{FB}) were determined to be 0.02 V and 0.75 V below the CB edges in the COFs.

Supplementary Figure 8. Ultraviolet photoelectron spectra (UPS) of PHT and PYT. Upper scale is in eV versus vacuum level (physical scale), bottom scale is in V versus SHE (electrochemical scale).

Figure 2b was updated using VB energy levels obtained from the UPS as the reference points followed by addition of optical gap values obtained from Tauc plots.

Figure 2g was updated as well.

Mott-Schottky plots were updated using 5, 10 and 15 kHz frequencies.

Supplementary Figure 9. Mott-Schottky plots. a) PHT. b) PYT.

6. In figure 3a, the insert text “PYT/anisle” should be changed to “PYT/anisole”.

Response: We appreciate referee careful reading of our manuscript and pointing at typos. The Figure was replaced by the corrected one.

7. The format errors in the references sections should be carefully examined (i. e. ref 26).

Response: We revised reference section and corrected typos.

8. Covalent organic frameworks (COFs) generally refer to the one with well-defined crystalline

porous structures. Please carefully check that whether such kind of amorphous polymers could be defined as COFs.

Response: We agree with the referee that the term 'covalent organic frameworks' should be used for crystalline porous structures. However, looking into literature, vast majority of materials from this class are amorphous or nanocrystalline. For example, in *Adv. Mater.* 2020, 32, 1904433; *Adv. Funct. Mater.* 2020, 30, 2003761. To ensure that our work is properly indexed by search engines and to maximize its visibility we prefer to keep the term 'Covalent organic frameworks (COFs)' and 'Covalent triazine frameworks (CTFs)' when refer to the subclass of the materials.

Reviewer #2 (Remarks to the Author):

In this work Savateev et al. synthesize two new amorphous CFTs and they were used for photo catalysis, bromination of arenes. For the synthesis it is employed 2,6-pyridinedicarboxaldehyde (PYT) or an aryl aldehyde (PHT), demonstrating the importance of pyridinic nitrogen. Later, they carry out the characterization of the two materials, using the usual techniques for these materials, such as XPS, BET, or IR.

The materials have a very low porosity (104 and 27 m²g⁻¹), being amorphous, and limiting their use to other types of applications (H₂, CO₂, etc.).

Response: We appreciate referee's time evaluating our manuscript and comments that highlight both strong and weak aspects of our work.

To convince the referee that despite being amorphous, the synthesized materials are useful photocatalysts, we applied them in currently central area of research – dual photoredox catalysis. Namely, we investigated our conjugated organic polymers in Ni-catalyzed C-N cross-coupling. The optimal yield and selectivity toward cross-coupling product have been achieved under illumination the photocatalytic system with 625 nm photons. A new section was added to the manuscript.

2.4. COFs in dual Ni-photocatalytic C–N cross-coupling

C–C and C–heteroatom cross-coupling of arylhalides with secondary amines,^{1,14} alcohols,^{15,16,17} thiols,¹⁸ trifluoroborates¹⁹ and alkenes²⁰ respectively mediated by a combination of cheap Ni salts and heterogeneous photocatalysts free of platinum group metals offers a convenient and scalable approach for synthesis of value-added organic compounds. We have chosen pyrrolidine and 4-bromobenzonitrile **3a** as reaction partners, NiBr₂·glyme as the pre-catalyst and DABCO as the base.¹ Upon illumination of the reaction mixture with 400 nm photons, the dehalogenation product **4a** was obtained selectively with 39% and 100% yield in case of PYT and PHT respectively (Table 2, entries 1,2). Despite extended absorption, upon illumination with 465 nm photons PYT gave **4a** with 5% yield, but failed to produce **5a** (entry 3). PHT and PYTnc gave **5a** with 29% and 30% respectively, but the main product was **4a** (entry 4,5). Addition of phenantroline as a ligand to the system containing PHT led to selective formation of **4a** (entry 6). Without Ni catalyst, PHT gave dehalogenation product with 44%

yield (entry 7), while reaction did not proceed without photocatalyst (entry 8). Under illumination with 525 nm photons for 120 h, PHT gave **5a** with 76% yield (entry 9), while PYT – only product **4a** with 5% yield (entry 10). To suppress completely formation of the dehalogenation product **4a**, we switched to red (625 nm) photons. PHT gave **5a** with 68% yield (entry 11). PYT did not convert any of the reagents (entry 12), while PYTnc gave **5a** with 7% (entry 13). The yield of **5a** was further improved to 89% without compromising the selectivity by extending the reaction time under red light to 168 h (entry 14).

Table 2. Dual Ni-photocatalytic C-N coupling.^a

Entry	Photocatalyst	Light	4a (%)	5a (%)	Conversion (%)
1	PYT	400 nm	39	0	39
2	PHT	400 nm	100	0	100
3	PYT	465 nm	5	0	5
4	PHT	465 nm	71	29	100
5	PYTnc	465 nm	70	30	100
6 ^b	PHT	465 nm	100	0	100
7 ^c	PHT	465 nm	44	0	44
8 ^d	–	465 nm	0	0	0
9 ^e	PHT	525 nm	24	76	100
10 ^e	PYT	525 nm	5	0	5
11 ^e	PHT	625 nm	0	68	70
12 ^e	PYT	625 nm	0	0	0
13 ^f	PYTnc	625 nm	0	7	7

^a Reaction conditions: 4-bromobenzonitrile (9.1 mg, 0.05 mmol), pyrrolidine (7.4 μ L, 0.09 mmol), NiBr₂·glyme (0.8 mg, 0.0025 mmol), DABCO (12.3 mg, 0.11 mmol), N,N-dimethylacetamide (1 mL), 48 h. Condition were adapted from the reference ¹. Yield and conversion were determined by GC-MS.

^b 1,10-phenanthroline (0.0025 mmol) was added as a ligand.

^c without NiBr₂·glyme.

^d without photocatalyst.

^e The reaction mixture was irradiated with light for 120 h instead of 48 h.

^f The reaction mixture was irradiated with light for 168 h instead of 48 h.

Overall, PHT enables selective coupling of electron deficient arylhalides **3a-d** with 18-89% yields (Figure 5), while electron rich substrates **3e-g** failed to give the coupling products (Supplementary Figure 31), which is one of the existing challenges in transition-metal catalyzed cross-coupling reactions.^{21,22}

Figure 5. Scope of dual Ni-photocatalytic C–N cross coupling. Arylhalide (0.05 mmol), PHT (12 mg), pyrrolidine (7.4 μ L, 0.09 mmol), NiBr₂·glyme (0.8 mg, 0.0025 mmol), DABCO (12.3 mg, 0.11 mmol), N,N-dimethylacetamide (1 mL), Light 302 mW cm⁻², 168 h. Yield and conversion (in parentheses) determined via GC-MS. ^a from 1-Bromo-4-iodobenzene.

Supplementary Figure 31. A list of aryl bromides that did not give the C–N coupling products. The starting materials have been recovered. Condition were adapted from the reference ¹. Arylhalide (0.05 mmol), PHT (12 mg), pyrrolidine (7.4 μ L, 0.09 mmol), NiBr₂·glyme (0.8 mg, 0.0025 mmol), DABCO (12.3 mg, 0.11 mmol), N,N-dimethylacetamide (1 mL), Light 302 mW cm⁻², 168 h. Yield and conversion (in parentheses) determined via GC-MS.

The structure of COFs and their activity in the C-N cross-coupling has been discussed in section 2.5 as follows.

In case of C–N cross-coupling, analysis shown in Figure 6a did not give a trend likely because the reaction is mediated by Ni-catalyst, while COF in combination with light facilitate destabilization of Ni-intermediates postulated in the catalytic cycle either via electron or energy transfer.^{23,24} In particular, reductive elimination from Ni(II)-intermediate is endothermic,^{25,26} but becomes exothermic if Ni is in oxidation state III (Supplementary Figure 33).

Supplementary Figure 33. A tentative mechanism of Ni-dual photocatalytic C–N cross coupling mediated by PHT. Adapted from reference ²⁷. RE – denotes “reductive elimination”, OA – “oxidative addition”.

Several conclusions related to the band structure of COFs and their activity could be made taking into account mechanism of dual Ni-photocatalytic C–N cross-coupling suggested in the literature, which is based on one-electron transfer.²⁷ Overall, relative alignment of band edges in PHT, which upon photon absorption gives the excited state that is more oxidative and less reductive compared to PYT, is beneficial to generate tentative Ni(I) and Ni(III) species (Supplementary Note 6).

Supplementary Note 6

Stronger oxidation power of the PHT excited state, 1.55 V compared to 0.75 V vs SCE in PYT, is advantageous for one-electron oxidation of the Ni(II)-intermediate (Supplementary Figure 33). Indeed, VB levels in PHT, mpg-CN(reference ¹) and Ir[dF(CF)₃ppy]₂(dtbbpy)PF₆(reference ²⁷) are 0.45-0.75 V more positive than the VB level in PYT (Supplementary Figure 39).

Supplementary Figure 39. Positions of band edges in semiconductors and redox potentials of molecular sensitizers. Positions of band edges in PHT, PYT (this work) and mpg-CN(reference ¹) and redox potentials of *Ir(III)/Ir(II) and I(III)/Ir(II) couples (reference ^{28, 27}). Intraband states (IBS) in PHT and PYT are schematically depicted based on the data shown in Figure 3.

PHT, mpg-CN(reference ¹) and Ir[dF(CF)₃ppy]₂(dtbbpy)PF₆(reference ²⁷) gave the C–N coupling product, while PYT did not.

At the same time, moderately reducing power of PHT excited state, CB -1.17 V compared to -1.75 V vs SCE in PYT, is sufficient to enable one-electron reduction of Ni(I) intermediate as the primary pathway for catalyst turnover.²⁷ The undesirable dehalogenation process, which typically requires strongly reductive photocatalyst excited state,^{29,30,31} is mitigated.

On the other hand, as inferred from transient absorption spectroscopy, the lifetime of the excited states exceeds >50 μ s for all COFs studied herein (Supplementary Note 3). Therefore, energy transfer pathway cannot be excluded.^{32,33}

Earlier Pieber et al. have shown that wavelength of incident photons is one of the leverages to decrease the rate of reductive elimination in the catalytic cycle and therefore mitigate deactivation of Ni-catalyst.¹⁴ Our results, presented in Table 2, indicate that by decreasing the energy of incident photons, we effectively suppress undesirable dehalogenation process and improve selectivity towards the product of C–N cross-coupling.

The optical properties obtained are very well performed, using different complementary techniques for the characterization of both materials, and showing that the incorporation of pyridine has a substantial impact on the photoredox properties.

In the last part of the work, the authors study the bromination of aromatic rings, where the only advantage in the use of the material is the in situ generation of the electrophilic species HOBR.

The material does not have any impact on the reactivity of the substrates because it is just depends on the HOBr that is generated in the reaction media.

Indeed, there is a final discussion of the structure activity of COF. The reactivity is occurring near the surface of the material or even in the reaction media. Therefore, the only impact of the material in the amount of the HOBr that the CTF is able to produce.

Response: In order to check whether there are any differences in the yields of the products using non-photocatalytic oxidative halogenation (a mixture of H_2O_2 and HBr in dark) and photocatalytic oxidative halogenation (generation of H_2O_2 in situ, this work) we took substrates from the Table 2.

Supplementary Table 5. Photocatalytic oxidative bromination with PYT versus oxidative bromination of electron rich aromatic compounds using a mixture of H_2O_2 and HBr in dark.

Photocatalytic oxidative bromination with PYT ^a					Oxidative bromination in dark ^b	
Entry	Substrate	Product	Time, h	Yield, %	Product	Yield (Conversion), % ^c
1			48	97 ^c		89(100)
2			72	80 ^c		75(75)
3			72	85 (4:1) ^d		73(73)
4			48	95 ^c		74(74)
5			24	75 ^c		69(69)
6			24	81 (1:1) ^c		74(74)

^a Reaction conditions: substrate (0.6 mmol), HBr (0.6 mL, 48 wt. %), MeCN (3 mL), PYT (4 mg), electron scavenger – O₂, blue LED module 2 (461 nm, 101 mW cm⁻²) for entry 1, 3-5 and white LED module (400-760 nm, 203 mW cm⁻²) for entry 2, at room temperature.

^b Reaction conditions: substrate (0.6 mmol), HBr (0.6 mL, 48 wt. %), H₂O₂ (10.6 mmol, 1.08 mL, 30 wt. %), MeCN (3 mL), at room temperature.

^c Isolated yields.

^d Yields determined by ¹H NMR with 1,4-dinitrobenzene as internal standard.

^e Yield and conversion (in parentheses) determined via GC-MS.

Using a mixture of H₂O₂ and HBr in dark, comparable yields of the brominated products have been obtained for the same substrates (Supplementary Table 5), which implies similar mechanism (see below).

We agree that there many other methods (also non-photocatalytic) that may be used to introduce halide atoms into electron rich aromatic compounds. In the context of this work, oxidative bromination is a convenient model reaction to correlated structure of the materials with their performance in photocatalysis.

Most (if not all) photocatalytic halogenations using both homogeneous and heterogeneous systems proceed in acidic medium. At the same time, under neutral conditions reaction scope is typically limited to only extremely electron-rich substrates. Our results also confirm this general observation. Supplementary Table 8 summarizes several articles and the role of acidic medium.

Supplementary Table 8. Role of acid in photocatalytic oxidative halogenation of electron rich aromatic compounds.

Entry	Photocatalyst	Role of acid	Reference
1	Microporous organic polymers (heterogeneous)	Participates in PCET to generate HO ₂ [•] from O ₂ .	7
2	 sodium anthraquinone-2-sulfonate (SAS, homogeneous)	Protonation of the photocatalyst increases excited oxidation potential of the photocatalyst, which in turn enables oxidation of thermodynamically more stable substrates via PCET. SAS-H [•] as the intermediate in the photocatalytic cycle is postulated.	8

3	 Riboflavin tetraacetate (RFT) (homogeneous)	Acetic acid is converted into peracetic acid in situ, which in turn enables oxidation of Cl ⁻ to OCl ⁻ .	9
4	Potassium poly(heptazine imide)	Participates in PCET to generate H ₂ O ₂ from O ₂ .	10
5	 [Acr-Mes]⁺ClO₄⁻	Participates in PCET to generate HO ₂ [•] from O ₂ .	11

Our COFs, PYT and PHT, have different content of pyridinic nitrogen atoms. Therefore, they are excellent models to address the question ‘What is the role of H⁺ in photocatalytic oxidative halogenation?’.

Analysis of our own results and results obtained by other research groups, point that in case of nitrogen-rich conjugated organic polymers, the role of proton in the photocatalytic mechanism is not only to participate in reduction of O₂ to H₂O₂ (or HO₂[•]) via PCET, protons in fact shift VB level to more positive values. Such conclusion is in agreement with the results obtained by Wu et al. for protonated g-C₃N₄.¹² Therein, protonation shifts the VB potential by 0.77V from 1.6 V to 2.37 V vs NHE.¹² Stabilization by 0.77 V also pulls down the potential of the CB, so that optical gap remains nearly constant. Protons (and in general cations) shift the CB and VB levels in nanoparticles of inorganic semiconductors, such as ZnO, to more positive values.¹³ Protonation of a photocatalyst surface increases oxidation power of the photogenerated holes compared to that under neutral pH.

Using structure of PYT as an example, schematically the mechanism of oxidative bromination of anisole is depicted in Supplementary Figure 38.

Supplementary Figure 38. Schematic mechanism of anisole oxidative bromination with PYT. In this figure only electron transfer is considered. Tentative local structure of PYT intermediates is shown. Upon excitation with light, electron is transferred from electron enriched part (*p*-phenylene units) to electron-deficient part (pyridine-2,6-diyl and/or triazine-linkers) of PYT.

Protonation of certain pyridine units in PYT gives [H-PYT]⁺Br⁻. Upon excitation, electron transfer from the VB, which is localized at *p*-phenylene moieties (the electron-enriched part of the polymer) to the CB, which is localized at protonated pyridine-2,6-diyl and/or triazine-linkers (the electron-deficient part of the polymer), gives excited state [H-PYT]⁺*Br⁻. One-electron oxidation of anisole followed by nucleophilic attack of bromide anion and abstraction of hydrogen atom by hydroperoxyl radical gives 4-bromoanisole and transient [H-PYT]*. Open-shell character of the tentative [H-PYT]* intermediate has been partially confirmed by increase of the amplitude of the EPR signal upon illumination of the mixture of PYT and anisole (Figure 4a). The latter transfers hydrogen atom to oxygen, which closes the photocatalytic cycle.

The suggested mechanism can rationalize the fact that in acidic environment we obtained 4-bromoanisole from anisole, despite its oxidation potential +1.81 V vs SCE² is more positive than the VB potentials of PHT (+1.55 V vs SCE) and PYT (+0.75 V vs SCE). Therefore, in acidic environment the VB potential of PHT is shifted by at least 0.26 V to more positive values. As a result, oxidation of anisole to the photocatalyst becomes feasible. Taking into account the band diagram of PYT, such shift is even larger, ca. 1.06 V, but due to higher content of pyridinic-nitrogen atoms (basic sites where protons reside) and 4 times larger surface area, is apparently plausible.

Section 2.5 has been extended to summarize these findings.

As evidenced by TR-PL data (Figure 2e, Supplementary Figure 32) excited state dynamics in PHT, PYT and PYTnc is different and depends strongly on the excitation wavelength. As deduced from Stern-Volmer plots shown in Figure 4e,f, electron transfer between PYT and anisole occurs at the rate $4.8 \cdot 10^9 \text{ M}^{-1} \text{ s}^{-1}$; between PYT and HBr – $1.6 \cdot 10^{10} \text{ M}^{-1} \text{ s}^{-1}$, which is limited by the diffusion of the reagents to the surface of the photocatalyst. Therefore, excitons that are separated with the rate smaller (or time greater) than the rate of electron transfer do not enable redox reaction. Figure 6a shows the correlation between the yield of 4-bromoanisole and the longest time ($\bar{\tau}_{max}$) of photogenerated charge carriers separation upon excitation at 470 and 640 nm. Therefore, the faster separation of the photogenerated charge carriers the higher is the yield of 4-bromoanisole under illumination with light of the corresponding wavelength. Despite higher activity of photocatalysts (higher yield of a product) in certain reactions is typically explained by improved absorption of photons of certain wavelength, our results indicate that absorption itself may not be the only factor. Despite PHT has significantly lower absorption at 620–625 nm compared to PYT (Figure 2a), it produces 4-bromoanisole with higher rate, 27% yield versus 12% (Table 1).

Potential of the occupied states, VB or intraband states laying close to the VB, defines the driving force for electron transfer from the substrate. Therefore, selective excitation of either $n-\pi$ or $\pi-\pi^*$ transitions in the conjugated polymers, such as carbon nitrides, can alter selectivity of a chemical reaction.^{34,35} Analysis of TCSPC data and steady-state emission spectra in combination with the results of COFs photocatalytic activity in oxidative bromination and dual Ni-photocatalytic C-N cross-coupling suggests that the concept is applicable for our systems as well. Thus, excitation of COFs with shorted wavelength, such as $\lambda \leq 450 \text{ nm}$ for PHT, $\lambda \leq 500 \text{ nm}$ for PYT and $\lambda \leq 650 \text{ nm}$ for PYTnc as deduced from Tauc plots (Supplementary Figure

7b), gives the excited state having $\pi-\pi^*$ character. On the other hand, excitation of COFs with longer wavelength, such as $\lambda > 450$ nm for PHT, $\lambda > 500$ nm for PYT and $\lambda > 650$ nm for PYTnc, gives the excited state having $n-\pi^*$ character. These excited states possess oxidation power that is defined by the band structure and the energy of incident photons (Relation 1).

Whenever a reaction occurs at the interface (e.g. heterogeneous catalysis), porosity plays a critical role. Some studies did not show a direct correlation between the degree of porosity and activity³⁶, while others reported a marked effect³⁷. In the context of oxidative bromination, an increase in surface area and pore volume in PYT may favor photocatalytic reaction by providing a larger number of basic sites for hosting protons, which facilitate oxidation of the substrates via multisite PCET – protons are supplied by HBr, while electrons by the substrate. Larger surface area and higher number of basic sites in PYT compared to PHT, facilitates trapping of electrons under ambient conditions as deduced from EPR spectra (Figure 2f). In photocatalytic cycle, compensation of the excessive negative charge on the surface of the photocatalyst by coupling it with H^+ stabilizes the system by ΔG_{st} , which for PYT can be as high as 1.06 eV when take into account oxidation potentials of the substrates from Figure 3 and VB potential of COFs (Figure 6b, Supplementary Figure 29). As a result, energy of photons at the PET step is directed towards the oxidation of thermodynamically more stable substrates.

Figure 6. Discussion of the structure-activity relationship. a) Correlation of the yield of 4-bromoanisole **2a** with the $\bar{\tau}_{max}$; b) Stabilization of the photocatalyst excited state by protonation of its surface allows utilizing energy at the PET step to oxidize thermodynamically more stable substrates. “R” stands for reagents, “P” – products.

In conclusion, although the work is technically very well performed, the reactivity explored with this material is not very new, nor is a very important advantage observed in the use of this material. In my opinion, it should be published in a more specific material journal.

Response: We appreciate referee pointing at technical quality of our work. In this round of revision we had a chance to look at our data from a different perspective. Comprehensive analysis enabled as to draw general conclusions, which are beyond just oxidative bromination of anisole and dual Ni-photocatalytic C-N cross-coupling (added during the revision), but allowed us to correlate structural features of COFs with their performance in photocatalysis, which is summarized in Conclusions:

Two amorphous COFs with extended light absorption in the visible range were prepared, instrumentalizing the gap between the conduction and valence band edges with the intraband states that originate from nitrogen lone pairs embedded into the framework. In PHT, which possesses p-phenylene linkers, the intraband states are located in the vicinity of the VB, while in PYT possessing pyridine-2,6-diyl linkers they are in the middle of the band gap. The intraband states are weakly emissive and photocatalytically active. Two reactions, oxidative bromination of electron rich aromatic compounds and dual Ni-photocatalytic C–N cross-coupling are mediated by the prepared COFs. In oxidative bromination, the intraband states allow the reaction to proceed under illumination with 625 nm photons. The yield of the products correlates with the time of photogenerated charge carriers separation. The role of acidic medium in oxidative halogenation has been explained by a stabilizing effect of H⁺ when coupled with the transfer of electron. Due to relative alignment of band edges in PHT that upon excitation gives more oxidative and less reductive excited state compared to PYT, COF mediates dual Ni-photocatalytic C–N cross-coupling of electron deficient arylhalides and pyrrolidine. Using 625 nm photons we suppressed completely undesirable dehalogenation process and maximized the yield of the cross-coupling product. The mechanism of photocatalytic reactions mediated by COFs is complex, but due to several features, such as relative alignment of band edges, presence of intraband states and an ensemble of basic pyridinic moieties, they offer several modes for substrate activation, which in homogeneous photocatalysis are typically achieved by combining several molecular mediators.

Taking into account generality and interdisciplinarity of our findings, we believe that our work merit publication in Nature Communications, a top scientific journal. It will guarantee high visibility of our work to the community working in photocatalysis, materials science and organic synthesis.

Sincerely,

Aleksandr Savateev

References for point-by-point replies

1. Ghosh I, Khamrai J, Savateev A, Shlapakov N, Antonietti M, König B. Organic semiconductor photocatalyst can bifunctionalize arenes and heteroarenes. *Science* **365**, 360-366 (2019).
2. Roth HG, Romero NA, Nicewicz DA. Experimental and Calculated Electrochemical Potentials of Common Organic Molecules for Applications to Single-Electron Redox Chemistry. *Synlett* **27**, 714-723 (2016).
3. Weinberg NL, Weinberg HR. Electrochemical oxidation of organic compounds. *Chemical Reviews* **68**, 449-523 (1968).
4. Savateev A, *et al.* Potassium poly(heptazine imide): transition metal-free solid-state triplet sensitizer in cascade energy transfer and [3+2]-cycloadditions. *Angew Chem Int Ed* **59**, 15061-15068 (2020).
5. Pavlishchuk VV, Addison AW. Conversion constants for redox potentials measured versus different reference electrodes in acetonitrile solutions at 25°C. *Inorganica Chimica Acta* **298**, 97-102 (2000).
6. Podgoršek A, Zupan M, Iskra J. Oxidative Halogenation with “Green” Oxidants: Oxygen and Hydrogen Peroxide. *Angewandte Chemie International Edition* **48**, 8424-8450 (2009).
7. Li R, *et al.* Photocatalytic selective bromination of electron-rich aromatic compounds using microporous organic polymers with visible light. *ACS Catal* **6**, 1113-1121 (2016).
8. Petzold D, König B. Photocatalytic Oxidative Bromination of Electron-Rich Arenes and Heteroarenes by Anthraquinone. *Advanced Synthesis & Catalysis* **360**, 626-630 (2018).
9. Hering T, Mühldorf B, Wolf R, König B. Halogenase-Inspired Oxidative Chlorination Using Flavin Photocatalysis. *Angewandte Chemie International Edition* **55**, 5342-5345 (2016).
10. Markushyna Y, *et al.* Halogenation of aromatic hydrocarbons by halide anion oxidation with poly(heptazine imide) photocatalyst. *Appl Catal B-Environ* **248**, 211-217 (2019).
11. Ohkubo K, Mizushima K, Iwata R, Fukuzumi S. Selective photocatalytic aerobic bromination with hydrogen bromide via an electron-transfer state of 9-mesityl-10-methylacridinium ion. *Chemical Science* **2**, 715-722 (2011).
12. Ye C, *et al.* Enhanced Driving Force and Charge Separation Efficiency of Protonated g-C₃N₄ for Photocatalytic O₂ Evolution. *Acs Catal* **5**, 6973-6979 (2015).
13. Valdez CN, Schimpf AM, Gamelin DR, Mayer JM. Proton-Controlled Reduction of ZnO Nanocrystals: Effects of Molecular Reductants, Cations, and Thermodynamic Limitations. *Journal of the American Chemical Society* **138**, 1377-1385 (2016).

14. Gisbertz S, Reischauer S, Pieber B. Overcoming limitations in dual photoredox/nickel-catalysed C–N cross-couplings due to catalyst deactivation. *Nat Catal* **3**, 611-620 (2020).
15. Malik JA, Madani A, Pieber B, Seeberger PH. Evidence for Photocatalyst Involvement in Oxidative Additions of Nickel-Catalyzed Carboxylate O-Arylations. *Journal of the American Chemical Society* **142**, 11042-11049 (2020).
16. Pieber B, *et al.* Semi-heterogeneous Dual Nickel/Photocatalysis using Carbon Nitrides: Esterification of Carboxylic Acids with Aryl Halides. *Angewandte Chemie International Edition* **58**, 9575-9580 (2019).
17. Zhao X, *et al.* Nickel-Coordinated Carbon Nitride as a Metallaphotoredox Platform for the Cross-Coupling of Aryl Halides with Alcohols. *Acs Catal* **10**, 15178-15185 (2020).
18. Cavedon C, Madani A, Seeberger PH, Pieber B. Semiheterogeneous Dual Nickel/Photocatalytic (Thio)etherification Using Carbon Nitrides. *Organic Letters* **21**, 5331-5334 (2019).
19. Khamrai J, Ghosh I, Savateev A, Antonietti M, König B. Photo-Ni-Dual-Catalytic C(sp²)–C(sp³) Cross-Coupling Reactions with Mesoporous Graphitic Carbon Nitride as a Heterogeneous Organic Semiconductor Photocatalyst. *Acs Catal* **10**, 3526-3532 (2020).
20. Khamrai J, Das S, Savateev A, Antonietti M, König B. Mizoroki–Heck type reactions and synthesis of 1,4-dicarbonyl compounds by heterogeneous organic semiconductor photocatalysis. *Green Chem* **23**, 2017-2024 (2021).
21. Qi Z-H, Ma J. Dual Role of a Photocatalyst: Generation of Ni(0) Catalyst and Promotion of Catalytic C–N Bond Formation. *Acs Catal* **8**, 1456-1463 (2018).
22. Kawamata Y, *et al.* Electrochemically Driven, Ni-Catalyzed Aryl Amination: Scope, Mechanism, and Applications. *Journal of the American Chemical Society* **141**, 6392-6402 (2019).
23. Zhu C, Yue H, Jia J, Rueping M. Nickel-Catalyzed C-Heteroatom Cross-Coupling Reactions under Mild Conditions via Facilitated Reductive Elimination. *Angewandte Chemie International Edition* **60**, 17810-17831 (2021).
24. Welin ER, Le C, Arias-Rotondo DM, McCusker JK, MacMillan DWC. Photosensitized, energy transfer-mediated organometallic catalysis through electronically excited nickel(II). *Science* **355**, 380-385 (2017).
25. Tasker SZ, Standley EA, Jamison TF. Recent advances in homogeneous nickel catalysis. *Nature* **509**, 299-309 (2014).
26. Zhu C, Yue H, Nikolaienko P, Rueping M. Merging Electrolysis and NickelCatalysis in Redox NeutralCross-Coupling Reactions: Experiment and Computation forElectrochemically Induced C–P and C–Se Bonds Formation. *CCS Chem* **2**, 179-190 (2020).

27. Corcoran EB, *et al.* Aryl amination using ligand-free Ni(II) salts and photoredox catalysis. *Science* **353**, 279-283 (2016).
28. Lowry MS, *et al.* Single-Layer Electroluminescent Devices and Photoinduced Hydrogen Production from an Ionic Iridium(III) Complex. *Chem Mater* **17**, 5712-5719 (2005).
29. Ghosh I, Ghosh T, Bardagi JI, König B. Reduction of aryl halides by consecutive visible light-induced electron transfer processes. *Science* **346**, 725-728 (2014).
30. Ghosh I, König B. Chromoselective photocatalysis: controlled bond activation through light-color regulation of redox potentials. *Angew Chem Int Ed* **55**, 7676-7679 (2016).
31. Marin M, Miranda MA, Marin ML. A comprehensive mechanistic study on the visible-light photocatalytic reductive dehalogenation of haloaromatics mediated by Ru(bpy)₃Cl₂. *Catalysis Science & Technology* **7**, 4852-4858 (2017).
32. Kudisch M, Lim C-H, Thordarson P, Miyake GM. Energy Transfer to Ni-Amine Complexes in Dual Catalytic, Light-Driven C–N Cross-Coupling Reactions. *Journal of the American Chemical Society* **141**, 19479-19486 (2019).
33. Das S, *et al.* Photocatalytic (Het)arylation of C(sp³)–H Bonds with Carbon Nitride. *Acs Catal* **11**, 1593-1603 (2021).
34. Kroutil W, *et al.* Chromoselective photocatalysis enables stereocomplementary biocatalytic pathways. *Angew Chem Int Ed* **n/a**, (2021).
35. Markushyna Y, Schüßlbauer CM, Ullrich T, Guldi DM, Antonietti M, Savateev A. Chromoselective Synthesis of Sulfonyl Chlorides and Sulfonamides with Potassium Poly(heptazine imide) Photocatalyst. *Angewandte Chemie International Edition* **60**, 20543-20550 (2021).
36. Sprick RS, *et al.* Tunable organic photocatalysts for visible-light-driven hydrogen evolution. *J Am Chem Soc* **137**, 3265-3270 (2015).
37. Yang C, *et al.* Molecular engineering of conjugated polybenzothiadiazoles for enhanced hydrogen production by photosynthesis. *Angew Chem Int Ed* **55**, 9202-9206 (2016).

REVIEWER COMMENTS

Reviewer #1 (Remarks to the Author):

The reviewers's concerns had been nicely addressed. The revised manuscript are also remarkably improved. Thus publication of the current version on Nature Communications could be recommended.

Reviewer #2 (Remarks to the Author):

In this revised manuscript, the author has improved the quality of the previous version. However, in opinion of this referee the impact is very moderate for a journal such as Nat. Commun. These materials are already well characterized and also use in the photocatalytic field (see e.g. Materials Today Chemistry 20 (2021) 100475 and references cited in there), and the chosen reactions do not represent any photocatalytic synthetic advanced in the area. Therefore, this manuscript should be published in a more specialized journal. Some suggestions:

- The authors are not working with COFs, since COFs need to be crystalline and porous. The definition of CTF is correct and it should be used in the hole manuscript instead of COF.
- The authors should add the reaction conditions employed for the obtention of the materials in Scheme 1
- When the authors talk about TGA, they should specify the conditions of the analysis (for example, air atmosphere, N₂ atmosphere...)
- The authors affirm that an enhancement of the BET surface of the material would entail better catalytic outputs. However, their materials present quite low porosity. In addition, this affirmation is too general, and its demonstration seems quite difficult. In order to address this question, some size-discrimination studies would be useful. If it is a matter of accessibility and diffusion of the substrates, better catalytic outputs should be obtained when a material with increased porosity is employed.
- In Table 1, the authors should specify the reaction time.
- In the sentence “using KBr as bromide source is beneficial por bromination of...”, the word bromination is misspelled.
- The authors should check the possibility of disassembly/leaching of their catalyst under the experimental conditions of photocatalytic bromination since the presence of HBr can give rise to undesired disassembly processes. Although they have checked its stability by FTIR, I would recommend filtrating the reaction after a catalytic run, add some substrate and quantify the yield after a second catalytic run in the absence of catalyst. The use of TEM after catalysis would be also helpful.
- In Figure 3, the authors should add the reaction time for each substrate. Moreover, the scope reached for the photocatalytic oxidative bromination of aromatic compounds is too scarce for a Nature Communications article. The authors should try more challenging substrates such as electron-poor reagents, even if it is necessary to change the reaction conditions. A possible scalability experiment would be desirable, too.
- In Table 2, the authors should again include the reaction times. However, extremely long reaction

times are required to carry out the C-N coupling (up to 168 h). Several works have been recently published in heterogeneous photoredox-Ni catalysis area, for example *Angew.Chem.* 2021, 133,10915 – 10922; and *ACS Catal.* 2020, 11758–11767. The authors should discuss these references and compared them with their obtained results. They should also check the stability and leaching issues of their materials under the experimental conditions employed in the C-N coupling.

Point-by-point replies to referee's comments to the manuscript "Red Edge Effect and Chromoselective Photocatalysis with Amorphous Covalent Triazine-based Frameworks" (Manuscript ID: NCOMMS-21-07813A)

Referee's comments in black

Our replies in blue

Changes made in the manuscript and Supplementary Information highlighted with orange background

REVIEWER COMMENTS

Reviewer #1 (Remarks to the Author):

The reviewers's concerns had been nicely addressed. The revised manuscript are also remarkably improved. Thus publication of the current version on Nature Communications could be recommended.

Response: We appreciate referees' recommendation to publish the current version of the manuscript in Nature Communications.

Reviewer #2 (Remarks to the Author):

In this revised manuscript, the author has improved the quality of the previous version. However, in opinion of this referee the impact is very moderate for a journal such as Nat. Commun. These materials are already well characterized and also use in the photocatalytic field (see e.g. Materials Today Chemistry 20 (2021) 100475 and references cited inthere), and the chosen reactions do not represent any photocatalytic synthetic advanced in the area. Therefore, this manuscript should be published in a more specialized journal. Some suggestions:

Response: We appreciate that the referee acknowledged the improved quality of our manuscript. We would like to point out that the central aspect of our work is the chromoselective behavior of CTFs in two reactions. Chromoselective photocatalysis is the last frontier in photocatalytic research, which allows tuning selectivity of reactions employing photons of different wavelength. To this date, only a few articles have been published on this topic (Scopus, December 2021, search keyword "chromoselective"): [Angew. Chem. Int. Ed. 2016, 55, 7676-7679], [Nat Commun 2018, 9, 60], [Nature Commun. 2019, 10, 2634], [Nature Catal. 2020, 3, 611-620], [Angew. Chem. Int. Ed. 2021, 60, 20543-20550], [Angew. Chem. Int. Ed. 2021, 60, 6965-6969].

Most importantly, the dual Ni-photocatalytic cross-coupling is currently a central area of research, which is supported by a current Scopus analysis (search keywords "dual" AND "photoredox" AND "catalysis", December 2021):

- There are 415 articles in the output
- 25% of these articles are from top chemistry journals such as Nature journals, Science, Angewandte Chemie and Journal of the American Chemical Society.

In light of the aforementioned, we strongly believe that the impact of chromoselective and dual Ni-photocatalysis is far from being moderate. In fact, recent publications such as Nat Commun. 2018, 9, 60 and Nat Commun. 2019, 10, 2634 underline the interest in this topic.

In several articles on dual transition metal photocatalytic C-N coupling it has been pointed out that a side-reaction is reductive dehalogenation of arylhalides.[Nat Commun. 2021, 12, 1646],[Nat Catal. 2020, 3, 611] We also observed this side-reaction in our studies. But, we solved this issue by simply changing the light source from a blue-LED to a red-LED. Considering the simplicity of our successful approach, it is applicable to other photocatalytic systems.

The article mentioned by the referee and several references therein, such as J. Am. Chem. Soc. 2020, 142, 6856, Adv. Mater. 2012, 24, 2357, Angew. Chem. Int. Ed. 2008, 18, 3450, offer an explanation for the relatively low surface area of CTFs, that is, stacking of the neighbouring layers in an overall AB fashion. Nanocrystalline structure of the CTFs impedes confirmation of this hypothesis. References to these articles have been added to the relevant sections in our article:

Lower surfaces of the CTFs might stem from staking neighbouring layers in AB fashion. [J. Am. Chem. Soc. 2020, 142, 6856],[Adv. Mater. 2012, 24, 2357],[Angew. Chem. Int. Ed. 2008, 18, 3450]

By virtue of the nanocrystalline nature of CTFs, a precise determination of the stacking type, that is, AA versus AB, is impossible.

The aforementioned article contains 34 references, three of which have been added to our manuscript as relevant to the topic.

- The authors are not working with COFs, since COFs need to be crystalline and porous. The definition of CTF is correct and it should be used in the hole manuscript instead of COF.

Response: As suggested by the referee we have changed the name of materials from COFs to CTFs.

- The authors should add the reaction conditions employed for the obtention of the materials in Scheme 1

Response: Synthesis conditions have been added to Scheme 1.

- When the authors talk about TGA, they should specify the conditions of the analysis (for example, air atmosphere, N₂ atmosphere...)

Response: Conditions under which TGA analysis was conducted have been added to the Supplementary Information

3.8 Thermal gravimetric analysis (TGA)

TGA measurement was performed using a thermo microbalance TG 209 F1 Libra coupled with a Thermostat Mass spectrometer (Pfeiffer Vacuum) with an ionization energy of 75 eV. Analysis was conducted under N₂ atmosphere.

and manuscript.

Thermal gravimetric analysis (TGA) measurements performed under a N₂ atmosphere reveal that PHT and PYT are stable up to 300 °C without significant loss of mass (around 10%) (Supplementary Figure 6).

- The authors affirm that an enhancement of the BET surface of the material would entail better catalytic outputs. However, their materials present quite low porosity. In addition, this affirmation is too general, and its demonstration seems quite difficult. In order to address this question, some size-discrimination studies would be useful. If it is a matter of accessibility and diffusion of the substrates, better catalytic outputs should be obtained when a material with increased porosity is employed.

Response: The results obtained in a set of photocatalytic experiments clearly demonstrate that porosity is not a primary factor that defines performance of the CTFs.

However, porosity is neither the only and nor the primary factor that determines the CTF performance. If this would be the case, due to, for example, improved accessibility of the substrates, PYT, which features a 4 times larger surface area and 6 times larger cumulative pore volume than PHT, would be more active in all of the photocatalytic experiments. Moreover, PYT would be more active than PHT regardless of the photon wavelength, which was selected for the photocatalytic experiments. This is, however, not the case (Supplementary Figure 34). In fact, the CTF activity correlates with its surface area and porosity only in the oxidative halogenation of anisole under 468 nm illumination. But, PHT, which has a lower surface area as well as lower porosity, is more active in oxidative halogenation under 625 nm illumination and dual Ni-photocatalytic C–N coupling under 465, 525, and 625 nm illumination.

Supplementary Figure 34. Correlation between the surface area and cumulative pore volume of the CTFs and yield of 4-bromoanisole **2a** and the product of C–N cross-coupling **5a**. Grey bars label data points obtained for PHT, pink bars – PYT.

- In Table 1, the authors should specify the reaction time.

Response: The reaction time is specified in Table 1.

Table 1 Screening of reaction conditions.

Entry	Photocatalyst	Time (h)	Light	Yield (%) ^a
1	PYT (4 mg)	4	468 nm ^b	82
2	PHT (4 mg)	4	468 nm ^b	7
3	PYTnc (4 mg)	4	468 nm ^b	100
4	PYT (4 mg)	24	468 nm ^b	99
5	PHT (4 mg)	24	468 nm ^b	56
6	-	24	468 nm ^b	2
7	PYT (4 mg)	24	-	7
8 ^c	PYT (4 mg)	24	468 nm ^b	97
9 ^d	PYT (4 mg)	24	468 nm ^b	99
10 ^e	PYT (4 mg)	24	468 nm ^b	98
11 ^f	PYT (4 mg)	24	468 nm ^b	96
12 ^g	PYT (4 mg)	24	468 nm ^b	95
13	PHT (4 mg)	24	625 nm ^h	27
14	PYT (4 mg)	24	625 nm ^h	12
15	PYTnc (4 mg)	24	625 nm ^h	75

Reaction conditions: photocatalyst (4 mg); anisole (0.02 mmol); HBr (0.1 mL, 48 wt. %); MeCN (0.5 mL); electron scavenger – O₂; at room temperature.

^aYields estimated by ¹H NMR with 1,4-dinitrobenzene as internal standard. Exemplary NMR spectrum of the reaction mixture is shown in Supplementary Figure 22.

^bBlue LED module 1 (468 nm, 14 mW cm⁻²)

^cReaction with 0.02 mL of HBr (48 wt. %).

^dReaction with 0.05 mL of HBr (48 wt. %).

^eSecond run.

^fThird run.

^gFourth run.

^hRed LED module (625 nm, 302 mW cm⁻²)

- In the sentence “using KBr as bromide source is beneficial for bromination of...”, the word bromination is misspelled.

Response: We appreciate referee reading the manuscript carefully. The typo has been corrected.

Using KBr as bromide source is beneficial for bromination of electron rich acidophobic compounds, such as 3,4-ethylenedioxythiophene **1g**, which gave 2,5-dibromo-3,4-ethylenedioxythiophene **2g** in 44% yield (Supplementary Table 9).

- The authors should check the possibility of disassembly/leaching of their catalyst under the experimental conditions of photocatalytic bromination since the presence of HBr can give rise to undesired disassembly processes. Although they have checked its stability by FTIR, I would recommend filtrating the reaction after a catalytic run, add some substrate and quantify the yield after a second catalytic run in the absence of catalyst. The use of TEM after catalysis would be also helpful.

Response:

Results of control experiments to check for “leaching” suggests the lack of any organics derived from PYT, which are soluble in acetonitrile, and, which could act as sensitizers in the oxidative bromination of anisole (Supplementary Table 6).

Supplementary Table 6. Control experiments for leaching of photocatalytically active organic moieties from PYT.

Entry	Yield (%)	Conversion (%)
1 ^a	99	100
2 ^b	0	0

^a Reaction conditions: anisole (0.6 mmol); HBr (0.6 mL, 48 wt. %); MeCN (3 mL); PYT (4 mg); electron scavenger – O₂; at room temperature; 48 h; 461 nm (101 mW cm⁻²).

^b After the photocatalytic experiment performed according to the conditions specified in entry 1, PYT was separated from the reaction mixture by centrifugation at 13000 rpm. The solution was loaded into the clean photoreactor. A new portion of anisole (0.6 mmol) and HBr (0.6 mL, 48 wt. %) was added to the solution. The solution was stirred under irradiation with blue LED (461 nm, 101 mW cm⁻²) for 48 h under atmosphere of O₂.

TEM would definitely be a technique of choice to characterize recovered after the photocatalytic experiment materials, if they were crystalline. The studied CTFs are nanocrystalline as indicated by powder XRD (Figure 2 and Scherrer equation in Supplementary Note 1). In addition, because of their sensitivity to high-energy electron beam (Supplementary Figure 4), the CTFs appear as amorphous materials. Finally, TEM is a technique suitable to characterize a local (nm range) structure of the materials. To characterize material as a whole, elemental analysis and in general techniques that sample data from a larger area/volume are more reliable, which are FT-IR spectroscopy (Supplementary Figure 17), steady-state (Supplementary Figure 19) and time-resolved fluorescence spectroscopy (Supplementary Figure 20) and DRUV-vis absorption spectroscopy (Supplementary Figure 18). These techniques clearly indicate that the structure of PYT was not altered significantly. In the DRUV-vis absorption spectrum (Supplementary Figure 18) of PYT recovered after the photocatalytic experiment, partial bleaching of the band responsible for n-π* transitions was observed. Such bleaching we attribute to partial protonation of the pyridinic lone pairs of the CTF structure.

In the revised version of the manuscript, we recovered PYT from the reaction mixture after photocatalytic oxidative halogenation of anisole and determined C, N, H content by combustion elemental analysis.

Supplementary Table 5. Combustion elemental analysis of fresh PYT and recovered after the photocatalytic oxidative bromination of anisole.

Entry	C (%)	N(%)	H (%)
PYT	56.8	19.2	4.2
PYT recycled	55.1±1	17.0±0.5	4.1±0.2

- In Figure 3, the authors should add the reaction time for each substrate. Moreover, the scope reached for the photocatalytic oxidative bromination of aromatic compounds is too scarce for a Nature Communications article. The authors should try more challenging substrates such as electron-poor reagents, even if it is necessary to change the reaction conditions. A possible scalability experiment would be desirable, too.

Response: Reaction time has been added after the yield of each compound in Figure 3.

Figure 3. Scope of the photocatalytic bromination of aromatic compounds using PYT as a photocatalyst. Reaction conditions: substrate (0.6 mmol); HBr (0.6 mL, 48 wt. %); MeCN (3 mL); PYT (4 mg); electron scavenger – O₂; at room temperature.

^a Isolated yields

^b Yields determined by ¹H NMR with 1,4-dinitrobenzene as internal standard. NMR spectra of reaction mixture are shown in Supplementary Figure S24-S29.

^c Reaction conditions: EDOT (0.2 mmol); KBr (119 mg, 1 mmol); DMSO-d₆:H₂O (1 mL, 9:1); PYT (12 mg); O₂ (1 bar); at room temperature.²

^d blue LED module 2 (461 nm, 101 mW cm⁻²)

^e white LED module (400-760 nm, 203 mW cm⁻²)

Scalability of the photocatalytic oxidative bromination with PYT was investigated on 6 mmol of anisole as described in the experimental procedure.

4.3 Method of photocatalytic oxidative bromination of aromatic compounds (6 mmol scale reaction)

A 3 neck custom made flask (100 mL) was charged with a mixture of anisole (652 μL, 6 mmol), HBr (6 mL, 48 wt. %), PYT (40 mg) and acetonitrile (30 mL). Magnetic stir bar was placed in the flask. Then, a cold finger attached to water cooling was immersed into the photoreactor. Temperature of the reaction mixture was maintained at 20-25 °C enabling the circulation water. The balloon filled with O₂ was attached to one of the necks. The pressure of O₂ during the experiment was maintained at ca. 1 bar. The reaction mixture was vigorously stirred for 48 h under illumination with 3 LED modules (emission maximum $\lambda = 465$ nm) with the distance between the photoreactor and LEDs of 1 cm (optical power supplied to the photoreactor by 3 LEDs was 1110 mW·cm⁻²). Yield and conversion were determined by ¹H NMR with internal standard.

Supplementary Figure 23. Oxidative bromination performed on 6 mmol scale of anisole.

The reaction may be further scaled up to 6 mmol of **1a**, while keeping the reaction time unchanged (48 h) via increasing the photon flux proportionally to the amount of the substrate (Supplementary Figure 23). In this case, the yield of **2a** was 87%.

We need to emphasize that the primary focus of our article is not developing a methodology for oxidative bromination of aromatic hydrocarbons, but *studying chromoselective behavior of the synthesized CTFs*.

As summarized in Supplementary Figure 30, the relative alignment of the VB in CTFs with the respect to the oxidation potential of a substrate defines the driving force for electron transfer.

Despite VB in PYT is less positive than the oxidation potential of most hydrocarbons shown in Supplementary Figure 30 (and Figure 3), it does enable oxidative bromination of the substrates, which we explained by the stabilizing effect of proton that comes from HBr (mechanism involves PCET, not pure ET). Despite PCET can extend the electrochemical window accessible with our CTFs, we are limited to substrates having oxidation potential < 1.81 eV, such as anisole.

Another issue that comes into play if we move from electron rich substrates to electron deficient ones, is the reduction potential of the latter shifts to more positive values. See, for instance, dependence of redox potentials of several classes of organic compounds on their electronic structure investigated in *Synlett* 2016 27 714-723. It means that in photoexcited CTFs, the photogenerated electron will be transferred from the CB to the electron-deficient substrate. It is especially true for PYT, which has CB potential of -1.75 V vs SCE. In other words, *CTFs will likely trigger reduction of electron deficient substrates instead of their oxidation.*

In addition, strongly-electron deficient substrates, such as nitro-substituted aromatic compounds absorb in visible range. Therefore, it is likely that reaction with such substrates will not be photocatalytic, but photochemical. Considering that the article focuses on studying chromoselective photocatalysis, we use substrates, which will not have compatibility issues with the photocatalytic system.

- In Table 2, the authors should again include the reaction times. However, extremely long reaction times are required to carry out the C-N coupling (up to 168 h). Several works have been recently published in heterogeneous photoredox-Ni catalysis area, for example *Angew.Chem.* 2021, 133,10915 –10922; and *ACS Catal.* 2020, 11758–11767. The authors should discuss these references and compare them with their obtained results. They should also check the stability and leaching issues of their materials under the experimental conditions employed in the C-N coupling.

Response: The column with reaction times was added to Table 2.

Table 2. Dual Ni-photocatalytic C-N coupling.^a

Entry	Photocatalyst	Light	Time, h	4a (%)	5a (%)	Conversion (%)
1	PYT	400 nm	48	39	0	39
2	PHT	400 nm	48	100	0	100
3	PYT	465 nm	48	5	0	5
4	PHT	465 nm	48	71	29	100
5	PYTnc	465 nm	48	70	30	100

6 ^b	PHT	465 nm	48	100	0	100
7 ^c	PHT	465 nm	48	44	0	44
8 ^d	–	465 nm	48	0	0	0
9	PHT	525 nm	120	24	76	100
10	PYT	525 nm	120	5	0	5
11	PHT	625 nm	120	0	68	70
12	PYT	625 nm	120	0	0	0
13	PYTnc	625 nm	168	0	7	7
14	PHT	625 nm	168	0	89	91
15 ^e	PHT	625 nm	168	0	80	80

^a Reaction conditions: photocatalyst (12 mg), 4-bromobenzonitrile (9.1 mg, 0.05 mmol), pyrrolidine (7.4 μ L, 0.09 mmol), NiBr₂·glyme (0.8 mg, 0.0025 mmol), DABCO (12.3 mg, 0.11 mmol), N,N-dimethylacetamide (1 mL), 48 h. Condition were adapted from the reference ². Yield and conversion were determined by GC-MS.

^b 1,10-phenanthroline (0.0025 mmol) was added as a ligand.

^c without NiBr₂·glyme.

^d without photocatalyst.

^e data obtained after using PHT for three consecutive rounds.

Longer reaction times are quite common, when photons of longer wavelengths are employed in the dual Ni-photocatalysis.[Nat Catal. 2020, 3, 611-620]

A set of techniques unambiguously confirmed that the chemical structure of PHT remained intact throughout the dual Ni-photocatalysis, which is explained by extremely mild conditions – low-energy electromagnetic radiation (625 nm) and 25 °C (Supplementary Note 6).

Supplementary Note 6

To check stability of PHT, we recovered the CTF and characterized by a series of techniques. FT-IR revealed that chemical structure of PHT was not altered (Supplementary Figure 46). Steady-state photoluminescence spectra of PHT acquired under several excitation wavelengths (Supplementary Figure 47), revealed that red edge effect was preserved. Combustion C, N, H analysis of fresh PHT and PHT recovered after dual Ni-photocatalytic C–N cross-coupling of pyrrolidine and 4-bromobenzonitrile revealed nearly identical chemical composition

(Supplementary Table 12). Moreover, C/N ratio in fresh PHT (3.81) was close to that in PHT recovered after the photocatalytic experiment (3.86).

Supplementary Figure 46. FTIR spectra of fresh PHT and PHT recovered after dual Ni-photocatalytic C–N cross-coupling of pyrrolidine and 4-bromobenzonitrile.

Supplementary Figure 47. PL spectra of PHT recovered after dual Ni-photocatalytic C–N cross-coupling of pyrrolidine and 4-bromobenzonitrile upon a range of λ_{exc} .

Supplementary Table 12. Combustion elemental analysis of fresh PHT and recovered after dual Ni-photocatalytic C–N cross-coupling.

Entry	C (%)	N(%)	H (%)
Fresh PHT	63.7	16.7	4.4
Recovered PHT	62.1±0.02	16.1±0.04	4.1±0.03

Similar to oxidative halogenation of anisole with PYT, a control experiment to check for “leaching” of organics from PHT that could sensitize the dual-Ni photocatalytic C–N cross-coupling was negative – neither **5a** was formed nor any conversion of **3a** took place.

After three rounds of use, PHT gave **5a** with 80% yield (entry 15).

The articles mentioned by the referee have been discussed in the following context:

C–C and C–heteroatom cross-coupling of arylhalides with secondary amines,^{[2],[70]} alcohols,^{[71],[72],[73]} thiols,⁷⁴[Angew.Chem. 2021, 133,10915–10922] trifluoroborates⁷⁶ and alkenes⁷⁷ respectively mediated by a combination of cheap Ni salts and heterogeneous photocatalysts free of platinum group metals offers a convenient and scalable approach for synthesis of value-added organic compounds.

Optimizing, for example, the relative distance between the sensitizer and the transition metal site emerged as a viable strategy to improve quantum efficiency of the reaction. This has been demonstrated for an Ir-sensitizer and a Ni-catalyst integrated into a soft polymer.[ACS Catal. 2020, 11758–11767]

Sincerely,
Aleksandr Savateev

10.01.2022

REVIEWERS' COMMENTS

Reviewer #2 (Remarks to the Author):

The authors have improved the technical part of the publication, but on the other hand the idea still lacks sufficient originality for a journal like Nature communications. The authors in the response to this referee arguing that a good number of reactions have been published through the use of similar materials in good articles (also in other articles of much less impact). I think that doing an analysis of Scifinder does not justify the underlying argument, which is that the use of these materials in photocatalysis is well known, and also, as the author indicates has been used many times. Additionally, the reactions used are model reactions, where there is no improvement in the conditions obtained with other photocatalytic systems. The paper from the technical point of view seems impeccable to me, but the originality of the idea, of the material itself, already previously described, and of model reactions, well known make me think that the work should be published in a more specialized journal in materials.

Please check the items below carefully and add a response in each row of the table to indicate the changes that you have made. Please also check through any additional marked-up edits we may have provided within the manuscript file.

Author information

Our guidance:

Your response:

Please review your complete author list to verify that it is complete and accurate. We ask that you consult with your coauthors to ensure that all names, affiliations, and titles are represented correctly. Note that if any authors are added or removed after this point then all authors will be requested to provide approval documentation that could potentially delay the production of your paper.	Authors list is complete and accurate. All names, affiliations and titles are represented correctly.
Please include a list of the corresponding authors and an e-mail address for each one in the final version of the manuscript file.	Corresponding author: Aleksandr Savateev (email: Oleksandr.savatieiev@mpikg.mpg.de)

Article structure

Our guidance:

Your response:

We can accommodate up to 10 display items (Figures or Tables) in the main article. Each Figure and Table must fit easily within an A4 page (210 x 297 mm). Please ensure that the number and size of your Figures and Tables fulfil these requirements to avoid any delay in the acceptance of your article.	There are 9 display elements in the manuscript. Each element fits A4 page at 100% scale.
Please ensure your main manuscript file includes the following sections, in this order: Title Author list Affiliations Abstract Introduction Results Discussion (optional) Results and Discussion (optional) Methods (including Data Availability, Code Availability and Statistics subsections where relevant) References Acknowledgements Author Contributions Statement Competing Interests Statement Tables Figure Legends/Captions (for main text figures) We do not edit Supplementary Information files; they will be uploaded with the published article as they are submitted with the	The manuscript file includes the section in the following order: Title Author list Affiliations Abstract Introduction Results Discussion Methods (including Data Availability statement) References Acknowledgements Author Contributions Statement Competing Interests Statement Tables Figure Legends/Captions (for main text figures) Supplementary Information file is uploaded as a single PDF.

final version of your manuscript. Any tracked changes should be removed from the file and the file should be provided as a PDF file. Supplementary Figures do not need to be provided separately.

Main text

Our guidance:

Your response:

A full Methods section, divided into subsections and subheadings, must be provided in the main manuscript. There is no word limit to this section.	Method section is included. Supplementary methods are in the the Supplementary Information file.
Please remove or rename the Conclusion heading, as the main text should only include the sections Abstract, Introduction, Results, optional Discussion and optional Methods. We also allow a combined Results and Discussion section.	Conclusion section was transferred to Discussion section
Please do not use italics, bold font, underlining or speech marks unless required for technical terms (in both the main text and the display items).	We did not use any of these in the main text and display items
Please make sure that mathematical terms throughout your manuscript and Supplementary Information (including in figures, figure axes, and legends) conform strictly to the following guidelines. Equations must be supplied in editable format, and not as images. Scalar variables (e.g. x , V , χ) must be typeset in italic, whereas multi-letter variables and functions (e.g. log) must be formatted in roman. Vectors (such as the wavevector k or the magnetic field vector B) must be typeset in bold without italics.	Mathematical terms conform the guidelines
Please label equations sequentially as (1), (2), (3), etc.	There are two relations in the manuscript labeled as (1) and (2)
Atomic orbital notations (sp, d, etc.) and corresponding XPS labels should be typeset in italics throughout the main text, figures and Supplementary Information, whereas all accompanying superscripts/subscripts should be typeset in roman font.	Atomic orbitals notations and XPS labels are written in italic.

Figures and Tables

Our guidance:

Your response:

Please see the guidelines linked below for detailed instructions about how your figures should be prepared. Following these instructions will reduce the chances of delays should we need to request replacement artwork from you at a later stage. https://www.nature.com/documents/NRJs-guide-to-preparing-final-artwork.pdf	Figures comply with the guidelines
Please note that schemes are not used; sequences of chemical reactions or experimental procedures should be submitted as figures, with appropriate captions.	There are only Figures and Tables in the manuscript
If bold/italic formatting in tables is necessary, please define its meaning in a table footnote	Bold/italic formatting in tables is not used

Please make sure that the terms 'atomic units (a. u.)' or 'arbitrary units (arb. units)' are appropriately used.	In the manuscript, only arbitrary units are used and they are denoted as 'arb. units'.
Any abbreviations, symbols or colours present in your figures must be defined in the associated legends.	All symbols are defined in legends.
Chemical structures in figures should be drawn using the Nature Chemistry template or its settings: http://www.nature.com/authors/guides/NR_chemdraw_stylesheet.cds Refer to the Nature Research Chemical Structures Guide for all details: https://www.nature.com/authors/guides/ChemStructureGuide.pdf	Chemical structures were formatted using Nature Chemistry template
In each Figure and Supplementary Figure where error bars are used, they must be defined.	Error bars are defined in the footnotes to Figures and Supplementary Figures.

Data and Code

Our guidance:

Your response:

Nature journals strongly support public availability of data and code. Please deposit the data and code used in your paper into a public data repository, or alternatively, present the data as Supplementary Information. If data can only be shared on request, please explain why in your Data Availability Statement, and also in the correspondence with your editor. Please note that for some data types, deposition in a public repository is mandatory. Any restrictions on sharing of these data types must be clearly indicated in the statement and discussed with the editor. More information on our data deposition policies and available repositories can be found here: https://www.nature.com/nature-research/editorial-policies/reporting-standards#availability-of-data	Data availability statement was added to the article file.
All published manuscripts reporting original research in Nature Portfolio journals must include a data availability statement, as a separate section before the References and under the heading 'Data Availability'. The data availability statement must make the conditions of access to the "minimum dataset" that are necessary to interpret, verify and extend the research in the article, transparent to readers. This minimum dataset may be provided through deposition in public community/discipline-specific repositories, custom proprietary repositories or general repositories like Figshare, Zenodo and Dryad. Providing large datasets in supplementary information is strongly discouraged and the preferred approach is to make data available in repositories. Scientific Data, a Nature Portfolio journal, maintains a list of approved and recommended data repositories to support researchers seeking suitable repositories for their data (https://www.nature.com/sdata/policies/repositories). The Data Availability Statement should also reference any source data published alongside the paper. If DOIs are provided, we also strongly encourage including these in the Reference list (authors, title, publisher (repository name), identifier, year). For clinical datasets or third party data, please ensure that the statement adheres to our policy (https://www.nature.com/nature-research/editorial-policies/reporting-standards#availability-of-data)	The following statement was added to the manuscript file: The datasets generated during and/or analysed during the current study are available from the corresponding author on reasonable request.

Please use the following template to provide all the information stated above: The XX data generated in this study have been deposited in the YY database under accession code ZZ [add hyperlink here]. The XX data are available under restricted access for {insert reason}, access can be obtained by {explain how}. The raw XX data are protected and are not available due to data privacy laws. The processed XX data are available at YY. The XX data generated in this study are provided in the Supplementary Information/Source Data file. The XX data used in this study are available in the YY database under accession code ZZ [Add hyperlink here].	
Please ensure your Methods section contains a Data Availability Statement as described above.	Methods section contains Data Availability Statement

Methods

Our guidance:

Your response:

Centrifugation speeds must be reported in x g.	Confirm
---	----------------

References

Our guidance:

Your response:

Footnotes are not used; please incorporate any footnote text into the main text of the manuscript and remove the citation and remember to renumber the reference list accordingly, making the corresponding changes to the call outs in the main text.	Footnotes are not used.
Supplementary References should appear at the end of the Supplementary Information file, and must be self-contained and numbered from 1. References mentioned in both the main text and the Supplementary Information should be part of both reference lists so that the Supplementary Information does not refer to the reference list in the main paper and vice versa.	Supplementary References appear at the end of the Supplementary Information file.

End matter

Our guidance:

Your response:

Please supply an "Author Contributions" section after the "Acknowledgements" section that refers to all authors. For more information on the Author Contributions statement, please refer to our authorship policy(https://www.nature.com/nature-research/editorial-policies/authorship), and to the following Nature Editorial: https://www.nature.com/articles/4581078a.	Authors Contributions sections was added after Acknowledgement.
Please thoroughly review our policy on Competing Interests (http://www.nature.com/authors/policies/competing.html) and include a detailed statement in your final manuscript file, and in our manuscript tracking system. Please ensure the statements are identical in both. Be specific about how each point stated relates to the research, list applicable author initials, and/or patent numbers. If there are no competing interests, a negative statement ("The authors declare no competing interests") must be included.	The statement was included in the final manuscript file.
Nature Portfolio defines Competing Interest (CI) as financial and non-financial interests (including but not limited to funding, employment, stocks, shares, patents, personal or professional relationships with individuals or institutions, and	

unpaid membership advocacy) that could be perceived to directly undermine the objectivity, integrity, and value of a publication, or could be seen as having an influence on the judgments and actions of authors with regard to objective data presentation, analysis, and interpretation. Please thoroughly review our policy on Competing Interests and include a detailed statement both in your final manuscript file and in our manuscript tracking system. Please ensure the statements are identical in both. Be specific about how each point stated relates to the research and list applicable author initials, and/or patent numbers. If there are no competing interests, a negative statement must be included. https://www.nature.com/nature-research/editorial-policies/competing-interests	Negative statement is included.
Please confirm that all relevant funding awarded to each author is described in the Acknowledgements section. List each grant number, followed by the initials of the author who received it.	Initials of grant recipients are included in the acknowledgement section.

Preparing your manuscript files

Our guidance:

Your response:

Unless otherwise stated please limit individual file sizes to approximately 30MB. We strongly encourage the use of repositories for large datasets or source data due to size considerations.	All files are < 30MB.
Please supply a brief (maximum 300 characters, including spaces) summary of the main findings of the paper to be used on our website and in our e-alerts. The summary should be written in the third person in language suitable for a broad audience. The summary may be edited by the editors prior to publication. Please provide this summary in your cover letter.	A brief summary is provided in the cover letter
To ensure maximum visibility for your work, we may tweet about your paper following publication. If you would like us to include the Twitter handles of the first author(s), corresponding author(s), lab or institution in this tweet, please provide them in your cover letter. We would also welcome your suggestions for hashtags to use when tweeting about the work.	Twitter handles are provided in the cover letter
Please provide figures as individual vector files with editable text. Acceptable file types for figures are .ai, .eps, .pdf or Chem Draw for fully editable vector-based art. For detailed guidance on figure preparation, see https://www.nature.com/documents/aj-artworkguidelines.pdf	Whenever possible figures are provided as vector-based art. The rest are provided as .tiff files in high resolution of 600 dpi.
The use or adaptation of previously published images is strongly discouraged. If this is unavoidable, please request the necessary rights documentation to re-use such material from the relevant copyright holders and return this to us when you submit your revised manuscript. Please check whether your manuscript or Supplementary Information contain third-party images, such as figures from the literature, stock photos, clip art or commercial satellite and map data. For more information on what constitutes ownership by a third party, please contact our Editorial Assistant at naturecommunications@nature.com	Manuscript and Supplementary Information file do not contain previously published images.

Forms to complete

Our guidance:

Your response:

Editorial Policy Checklist Please update and upload a final version of the Editorial Policy Checklist with your revised manuscript files. A blank Editorial Policy Checklist can be found via the link below. Note that this form is a dynamic 'smart pdf' and must be downloaded and completed in Adobe Reader. Please update your current checklist or download from: https://www.nature.com/documents/nr-editorial-policy-checklist.zip	Editorial policy checklist was uploaded

You will need to upload:

Editorial Policy Checklist	+Included
Completed Third Party Rights Table (if relevant)	+Not relevant
A point-by-point response to the reviewers' comments	+Included
A completed copy of this checklist	+Included
The main article file in Microsoft Word format - please supply a version with tracked changes and a version with tracked changes accepted	++Included
Separate Figure files	+Included
Inventory of Supporting Information	+Included
A Supplementary Information file	+Included
ChemDraw files	+Included